# Development of a p62 biodegrader for autophagy targeted degradation

Zacharias Thiel[1] ✉, David Marcellin[1], Carole Manneville[1], Benedikt Goretzki[1], Luca Egger [1], Rob Maher[2], Noémie Siccardi[1], Laura Torres[3], Alexandra Probst[4], Catrin S. Müller [1], Nathalie George[5], Markus Vogel[5], Sabine Sinterhauf[5], Alexandra Lavoisier[5], Ji-Young Choi[5], Laurianne Forcellino[5], Alexandro Landshammer [6], Patrick Hauck[1], Celine Be[1], Frédéric Villard[1], Sascha Gutmann [1], Marc Meyer[1], Felix Freuler[1], Alexandra Hinniger[1], César Fernández[1], Suzanne Chau[1], Maude Patoor[1], Gilles Sansig[1], Gabriel Mitchell [3] & Beat Nyfeler [1] ✉

Autophagy-based targeted degradation offers a powerful complement to proteasomal degradation leveraging the capacity and versatility of lysosomes to degrade complex cargo. However, it remains unclear which components of the autophagy-lysosomal pathway are most effective for targeted degradation. Here, we describe two orthogonal induced-proximity strategies to identify autophagy effectors capable of degrading organelles and soluble targets. Recruitment of autophagy cargo receptors, ATG8-like proteins, or the kinases ULK1 and TBK1 is sufficient to trigger mitophagy, while only autophagy cargo receptors capable of self-oligomerization degrade soluble cytosolic proteins. We further report a single-domain antibody against p62 and its use as a heterobifunctional degrader to clear mitochondria. Fusing the p62 single-domain antibody to PINK1 enables selective targeting of damaged mitochondria. Our study highlights the importance of avidity for targeted autophagy and suggests that autophagy cargo receptors are attractive entry points for the development of heterobifunctional degraders for organelles or protein aggregates.

Macroautophagy (hereafter called autophagy) is one of the principal mechanisms for the removal and recycling of intracellular components. In this highly conserved pathway, intracellular constituents that are modified with polyubiquitin, or other signaling molecules such as lipids or sugars, are recognized by autophagy cargo receptors such as p62 (encoded by SQSTM1), NDP52, NBR1, OPTN and more[1–5]. These autophagy cargo receptors oligomerize, sequestering the cargo, and further recruit ATG8 family proteins (MAP1LC3A, MAP1LC3B, MAP1LC3C, GABARAP, GABARAPL1, and GABARAPL2) by binding via LC3-interacting regions (LIRs). The lipidated forms of the ATG8-family proteins are membrane-bound and contribute to the engulfment of cargo and the formation of double membrane vesicles, known as autophagosomes[6–8]. Several other factors such as ULK1 and FIP200 are co-recruited and facilitate the initiation and growth of autophagosomes[9–11] before they mature and ultimately fuse with lysosomes where the cargo is degraded by various hydrolytic enzymes, in an acidic environment. The wide range of enzymes present in lysosomes allows them to break down various intracellular

[1]Discovery Sciences, Biomedical Research, Novartis, Basel, Switzerland. [2]Discovery Sciences, Biomedical Research, Novartis, Cambridge, MA, USA. [3]Open Innovation at Global Health Disease Area, Biomedical Research, Novartis, Emeryville, CA, USA. [4]Global Health Disease Area, Biomedical Research, Novartis, Emeryville, CA, USA. [5]Biologics Research Center, Biomedical Research, Novartis, Basel, Switzerland. [6]Disease Area X, Biomedical Research, Novartis, Basel, Switzerland. ✉e-mail: zacharias.thiel@novartis.com; beat.nyfeler@novartis.com

macromolecules, such as proteins, nucleic acids, lipids, and carbohydrates. Furthermore, the size and high capacity of autophagosomes allow the engulfment of protein aggregates, organelles, and intracellular pathogens. These features of autophagy have spurred the development of modulators to enhance or block the pathway as potential therapeutics[12]. A prominent example is rapamycin that activates autophagic flux via inhibition of mTOR, a regulator of the autophagy initiation complex and of lysosomal biogenesis[13–15]. More recently, the modulation of autophagy has been extended to the cargo recognition step, enabling to selectively degrade targets[16].

Proteasome-based targeted protein degradation (TPD) has emerged as a promising therapeutic modality to remove disease drivers, and several therapies have reached the clinical stage[17]. TPD relies on heterobifunctional molecules (PROTACs), molecular glues or multivalent single-domain antibodies to induce proximity between a target and an effector protein, usually an E3 ligase. Proximity of the two interaction partners leads to polyubiquitination of the target followed by degradation by the proteasome[17–19]. Recently, alternative strategies that rely on targeted degradation via the autophagy lysosome pathway have been described[20]. These approaches expand the substrate scope to protein aggregates and entire organelles, cargos that are less accessible to degradation via the proteasome. Among the autophagy targeting effectors are autophagy cargo receptors (AUTOTACs[21]), ATG8-family proteins (ATTECs[22]), kinases (ULKRECs[23]) and unknown factors (AUTACs[24]). Despite the existence of several autophagy targeting degradation strategies, it remains unclear which part of the autophagy machinery is most efficacious to induce proximity-dependent clearance.

In this work, we establish a platform to identify autophagy effectors for proximity-induced degradation of organelles and soluble proteins. We show that induced proximity of different effectors such as autophagy cargo receptors, ATG8-like proteins or the kinases ULK1 and TBK1 are sufficient to trigger mitophagy. In contrast, self-oligomerizing autophagy cargo receptors outperform ATG8-like effectors and autophagy-related kinases in clearing a soluble cytosolic protein. Using this information, we develop a p62-targeting biodegrader and demonstrate its use for targeted mitophagy, including the selective degradation of damaged mitochondria.

## Results

### An induced proximity screen identifies several effectors that are sufficient to trigger mitophagy

To identify candidates for targeted autophagy degradation, we evaluated effectors that can clear mitochondria in a proximity-dependent manner. As an organelle, mitochondria represent a large and complex target that is inaccessible for degradation via the ubiquitin proteasome system. Lysosomal delivery of mitochondria was monitored in HeLa cells expressing the green fluorescent protein (GFP) and mCherry (mCh) tandem tag fused to the FIS1 mitochondrial targeting sequence, which tethers the reporter to the outer mitochondrial membrane (OMM) (mito-mCh-GFP)[25]. When mitochondria are trafficked to the lysosomes, the fluorescence of GFP is quenched in the acidic environment whereas mCh remains fluorescent. This reporter system allows to monitor mitophagy via the detection of mCh⁺/GFP⁻ punctae by fluorescence microscopy (Fig. 1A). Proximity to mito-mCh-GFP was induced by expressing autophagy effectors as N-terminal fusions with an anti-GFP single-domain antibody (VHH$_{GFP}$), which bind with high affinity to GFP[26]. The use of VHH$_{GFP}$ as a tool to induce proximity has been previously described in the context of targeted degradation approaches[27–30].

Using this assay, we tested 93 effector candidates such as E3 ligases, autophagy cargo receptors, ATG8-family members, and additional autophagy-related kinases and proteins. For the E3 ligases, we prioritized candidates with reported links to autophagy, with a focus on ligases that catalyze the formation of Lys63-linked polyubiquitin chains[31–33]. The BTB domain of the E3 ligase SPOP was included as control for proteasome-dependent targeted degradation[28], whereas unconjugated VHH$_{GFP}$ served as negative control. This screening platform identified 9 effectors that induce delivery of the mitochondrial reporter to lysosomes as indicated by a significant increase of mCh⁺/GFP⁻ punctae (Fig. 1B and Supplementary Dataset 1). These effectors represent autophagy cargo receptors (p62, NBR1, and NDP52), members of the ATG8 family (GABARAP, GABARAPL2, MAP1LC3A, and MAP1LC3B) and two kinases (TBK1 and ULK1). Interestingly, induced proximity with these effectors first caused a partial collapse of the mitochondrial network, followed by the formation of mCh⁺/GFP⁻ punctae 10–15 h post induction of the fusion constructs (Supplementary Fig. S1). Lysosomal targeting of the VHH$_{GFP}$ fusion constructs was validated by immunofluorescence-based co-localization with the lysosomal marker LAMP1 (Supplementary Fig. S2A). Furthermore, formation of mCh⁺/GFP⁻ punctae was sensitive to treatment with the lysosomal V-ATPase inhibitor Bafilomycin A1 as well as the VPS34 inhibitor PIK-III[34] (Supplementary Fig. S2B–D). Upon CRISPR knock out (K.O.) of ATG5, expression of VHH$_{GFP}$-p62 was no longer able to induce mCh⁺/GFP⁻ punctae, in line with lysosomal delivery via autophagy (Supplementary Fig. S2E, F). The mCh⁺/GFP⁻ punctae formation did not correlate with the expression levels of the VHH$_{GFP}$ fusion candidates (Fig. 1B). Expression of VHH$_{GFP}$ alone did not alter the mito-mCh-GFP morphology while SPOP resulted in complete loss of both mCh and GFP fluorescence, in line with proteasomal degradation of the reporter (Fig. 1C). In addition to the increased mCh⁺/GFP⁻ punctae phenotype, we observed that some effectors prompt the formation of large mCh⁺/GFP⁺ structures (exemplified by FIP200, Fig. 1C and Supplementary Fig. S1), suggesting that collapse of the mitochondrial network alone is not sufficient to induce mitophagy.

To test if lysosomal delivery of the mito-mCh-GFP reporter results in degradation of mitochondrial components, we assessed levels of the mitochondrial proteins COX4, TOMM20 and VDAC1 by immunoblot. TOMM20 and VDAC1 are transmembrane proteins located to the OMM, whereas COX4 localizes to the inner mitochondrial membrane. Induced proximity with the autophagy effectors led to a reduction of all three mitochondrial markers, suggesting robust clearance of mitochondrial content by mitophagy (Fig. 1D). In contrast, the E3 ligase SPOP selectively reduced the mito-mCh-GFP reporter without modulating levels of TOMM20, VDAC1, or COX4. Furthermore, expression of VHH$_{GFP}$-FIP200 did neither clear mito-mCh-GFP nor mitochondrial proteins, in accordance with the observation that collapse of the mitochondrial network alone is insufficient to induce mitophagy. In summary, our VHH$_{GFP}$-based screen suggests that induced-proximity with the autophagy cargo receptors p62, NBR1, and NDP52, the ATG8 family members GABARAP, GABARAPL2, MAP1LC3A, and MAP1LC3B, or the kinases ULK1 and TBK1 is sufficient to trigger targeted degradation of mitochondria.

### A small molecule-based induced proximity assay validates effectors for targeted autophagy degradation

To validate autophagy effectors identified from the cDNA screen with an orthogonal approach, we developed an induced proximity assay with a small molecule heterobifunctional dimerizer. We focused on two effectors from each protein class: the cargo receptors p62 and NBR1, the ATG8-like proteins GABARAP and MAP1LC3B, and the kinases ULK1 and TBK1. For this purpose, we expressed the autophagy effectors as N-terminal fusion proteins with the self-labeling protein HaloTag[35] in HeLa cells stably expressing mito-mCh-GFP fused to the self-labeling protein SNAPtag[36] (Fig. 2A) or by transient expression in Huh7 cells (Supplementary Fig. S3A, B). Treatment with the heterobifunctional compound HaXS8[37], a HaloTag-SNAPtag dimerizer, led to formation of mCh⁺/GFP⁻ punctae in a dose-dependent manner, reaching a full response at 10–100 nM (Fig. 2B). Induced proximity with autophagy cargo receptors (p62, NBR1) and ATG8 family members (GABARAP and MAP1LC3B) resulted in the strongest formation of

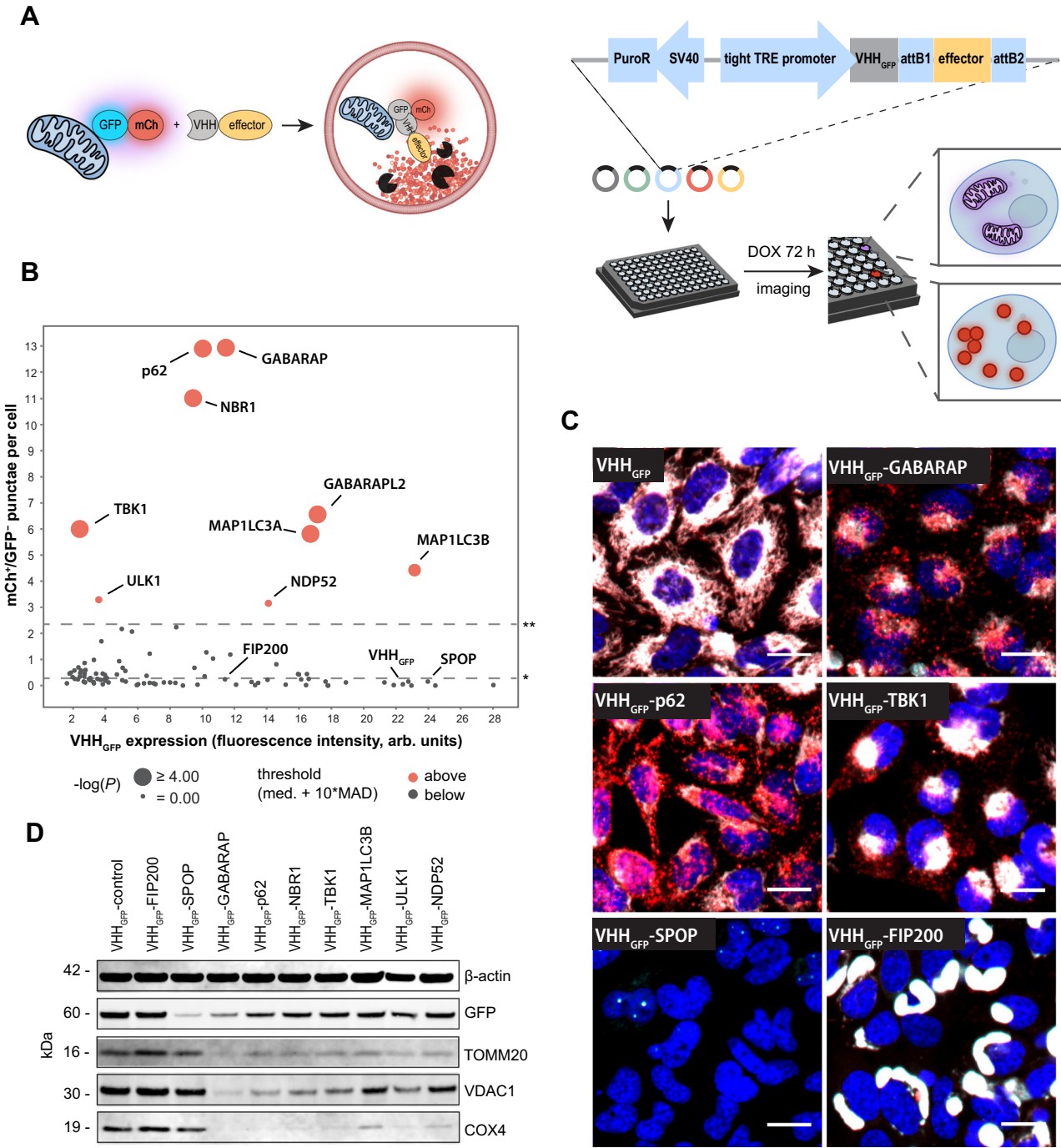

**Fig. 1 | An induced proximity cDNA screen to identify effectors for autophagy targeted degradation. A** Scheme of the VHH_GFP-based approach used to screen 93 autophagy effector candidates for proximity-dependent mitophagy. **B** Quantification of mCh+/GFP− granules per cell detected by fluorescence microscopy in stable HeLa mito-mCh-GFP cells with inducible expression of VHH_GFP-effector constructs 72 h post doxycycline (DOX) induction. Expression levels were determined by immunofluorescence staining against VHH after fixation. The data are shown as mean from $n = 3$ independent experiments. Statistical analysis was performed using ordinary one-way ANOVA with multiple comparison of each data point against the VHH_GFP using Dunnett's multiple comparisons test. *median; **median + 10 × median absolute deviation. Hits selected for further analysis are shown in red. **C** Representative confocal fluorescence microscopy images of the different mito-mCh-GFP phenotypes caused by induced proximity with autophagy effectors. Red = mCh, cyan = GFP, blue = nuclei. Scale bars are 30 μm. **D**. Immunoblot analysis of stable HeLa mito-mCh-GFP cells expressing VHH_GFP-autophagy effector fusions for 72 h.

mCh+/GFP− punctae, while the kinases ULK1 and TBK1 showed a partial response. Immunoblot analysis confirmed reduction of the mitochondrial proteins TOMM20, VDAC1 and COX4 when cells expressing autophagy effector-HaloTag fusions were treated with HaXS8 (Fig. 2C). Mitophagy was not triggered by HaXS8 alone as cells lacking HaloTag fusions did not show a reduction in mitochondrial proteins upon compound treatment. The immunoblot analysis of GFP as well as

HaloTag showed HaXS8-dependent high molecular weight bands, consistent with the formation of a ternary complex between the HaloTag fusions and mito-mCh-GFP-SNAPtag. All autophagy effector candidates showed remaining unbound HaloTag upon treatment with 10 nM HaXS8, indicating that the levels of autophagy effectors are not limiting. To confirm this, we performed a competition assay with HaloTag or SNAPtag labeled fluorophores to assess unoccupied

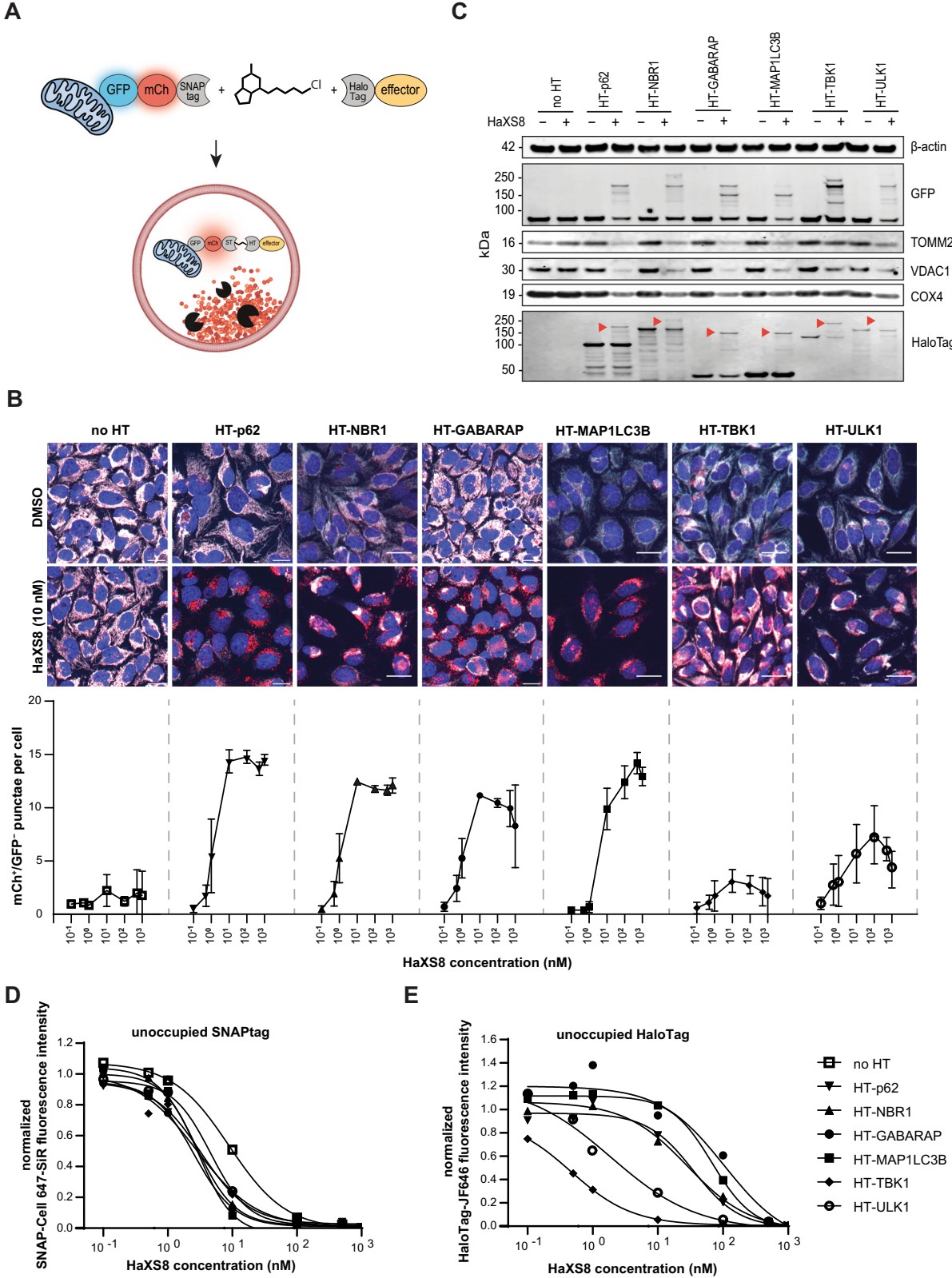

binding sites[38]. These experiments suggest that 10 nM HaXS8 almost saturates the SNAPtag on the mitochondrial reporter, whereas at this concentration most autophagy effectors showed remaining unbound HaloTag (Fig. 2D, E). These observations suggest that the differences in mitophagy observed among the tested autophagy effectors are unlikely to arise from variations in protein expression, but rather from differences in the effectiveness of the tested candidates.

## Self-oligomerizing autophagy cargo receptors outperform other autophagy effectors in the targeted degradation of a soluble protein

The induced proximity assays for mitophagy (Figs. 1 and 2) represent a paradigm where the target is locally concentrated at the mitochondrial membrane. To investigate the efficacy of the identified autophagy effectors in inducing targeted degradation of a soluble protein, we

**Fig. 2 | Validation of autophagy-effectors identified by the induced-proximity cDNA screen. A** Schematic representation of the approach used to induce VHH$_{GFP}$-independent proximity between mitochondria and autophagy effectors. Effectors are tethered to the surface of mitochondria by a small-molecule heterobifunctional dimerizer that links the self-labeling proteins HaloTag and SNAPtag. **B** Representative confocal fluorescence microscopy images and quantification of mCh$^+$/GFP$^-$ punctae in stable HeLa mito-mCh-GFP-SNAPtag cells with inducible expression of HaloTag fusions of autophagy effectors. Cells were treated with doxycycline and HaXS8 for 72 h. Red = mCh, cyan = GFP, blue = nuclei. Scale bars are 30 μm. The data are shown as mean and standard deviation from $n = 3$ independent biological replicates. **C** Immunoblotting of HeLa mito-mCh-GFP-SNAPtag cells with inducible expression of HaloTag fusions of autophagy effectors. Cells were treated with doxycycline and HaXS8 (0 or 10 nM) for 72 h. Red arrowheads indicate bands attributable to the ternary complex between mito-mCh-GFP-

SNAPtag, HaXS8, and HaloTag-autophagy effector. **D** Labeling of unoccupied SNAPtag in stable HeLa mito-mCh-GFP-SNAPtag cells with inducible expression of HaloTag autophagy effector fusions. Cells were treated with doxycycline and HaXS8 for 72 h followed by incubation with SNAP-Cell 647 SiR (200 nM) for 1 h. The mean fluorescence intensity of SNAP-Cell 647 SiR detected in cells was normalized to the signal measured in DMSO-treated cells. Data points were fitted using a non-linear, sigmoidal regression model. **E** Labeling of unoccupied HaloTag in stable HeLa mito-mCh-GFP-SNAPtag cells with inducible expression of HaloTag autophagy effector fusions. Cells were treated with doxycycline and HaXS8 for 72 h followed by incubation with HaloTag-JF646 (200 nM) for 1 h. The mean fluorescence intensity of HaloTag-JF646 detected in cells was normalized to the signal measured in DMSO-treated cells. Data points were fitted using a non-linear, sigmoidal regression model.

generated a HeLa cell line stably expressing cytosolic mCherry-GFP (cyt-mCh-GFP) and assayed lysosomal delivery upon expression of VHH$_{GFP}$-tagged effectors (Fig. 3A and Supplementary Fig. S4A–D). Upon induced proximity with p62 and NBR1, we observed robust formation of mCh$^+$/GFP$^-$ punctae and clearance of cyt-mCh-GFP (Fig. 3B–D). While proximity with GABARAP and MAP1LC3B resulted in a partial response, ULK1 and TBK1 were ineffective in clearing cyt-mCh-GFP. In accordance to the experiments performed with mito-mCh-GFP, mCh$^+$/GFP$^-$ punctae formation was inhibited, when cells were treated with Bafilomycin A1 or PIK-III (Supplementary Fig. S4D). Since both p62 and NBR1 contain a PB1 domain through which these cargo receptors homo- and hetero-oligomerize[2,39], we hypothesized that the oligomerization property of these effectors results in a superior efficacy to induce the targeted degradation of soluble targets. To test this hypothesis, we focused on p62 and introduced mutations or deletions in critical domains (Fig. 3E). Specifically, we evaluated the effect of preventing oligomerization of p62 by deleting (p62$_{103-440}$, hereafter called p62_ΔPB1) or mutating the PB1 domain (p62_PB1$_{mut}$, hereafter called p62_PB1$_{mut}$)[39–41]. In addition, p62$_{W338A/L341A}$ (hereafter called p62_LIR$_{mut}$) was tested to understand the contribution of ATG8 binding[42].

When VHH$_{GFP}$-p62_ΔPB1 and VHH$_{GFP}$-p62_PB1$_{mut}$ were expressed in HeLa cyt-mCh-GFP cells, we observed that mutations of the oligomerization domain completely abolished delivery of soluble cyt-mCh-GFP to lysosomes, indicated by the decrease of mCh$^+$/GFP$^-$ punctae formation and lack of protein degradation relative to canonical p62 (Fig. 3F, G and Supplementary Fig. S5A, B). Expression of VHH$_{GFP}$-p62_LIR$_{mut}$ led to the formation of large mCh$^+$/GFP$^+$ structures and significantly fewer mCh$^+$/GFP$^-$ punctae (Fig. 3F, G and Supplementary Fig. S5A, B). These data suggest that p62 can still engage a cytosolic target and oligomerize in the absence of binding to ATG8-like proteins but its ability to drive lysosomal delivery is reduced. We then tested the effects of these mutations on a scaffolded target. When expressed in HeLa mito-mCh-GFP cells, the PB1 mutants, VHH$_{GFP}$-p62_ΔPB1 and VHH$_{GFP}$-p62_PB1$_{mut}$, were still able to induce mitophagy (Fig. 3F, G and Supplementary Fig. S5A, B). Mutation of the LIR domain did not significantly alter the delivery of mitochondria to lysosomes, presumably due to co-recruitment of endogenous p62 by VHH$_{GFP}$-p62_LIR$_{mut}$. To test the contribution of co-oligomerization of the VHH$_{GFP}$-p62 mutants with endogenous p62, we tested these constructs in HeLa mito-mCh-GFP p62 K.O. cells (Supplementary Fig. S6). Expression of VHH$_{GFP}$-p62_ΔPB1 and VHH$_{GFP}$-p62_PB1$_{mut}$ led to the formation of mCh$^+$/GFP$^-$ punctae, even in the absence of endogenous p62, however to a lesser extent than in HeLa mito-mCh-GFP cells (Supplementary Fig. S6A–D). Expression of VHH$_{GFP}$-p62_LIR$_{mut}$ in p62 K.O. cell resulted in complete absence of formation of mCh$^+$/GFP$^-$ punctae, indicating that this construct mainly drives phagophore assembly via co-recruitment of endogenous p62.

To evaluate if oligomerization and binding of ATG8-like proteins are not only required but also sufficient for triggering autophagy

targeted degradation, we studied minimal constructs containing the p62 PB1 domain or an extended sequence of the p62 LIR motif (Fig. 3H)[41,42]. To increase avidity, we tested one to three repeats of the LIR motif as well as a combination of the PB1 domain and the LIR motif[43]. We expressed these minimal constructs in HeLa cyt-mCh-GFP as well as in HeLa mito-mCh-GFP cells and assessed lysosomal delivery and protein degradation (Fig. 3I, J and Supplementary Fig. S5C, D). For the cytosolic target cyt-mCh-GFP, one to three repeats of the LIR domain only minimally increased the formation of mCh$^+$/GFP$^-$ punctae. The combination of the PB1 domain and a single LIR peptide was as efficient as full length p62 in clearing cyt-mCh-GFP. Expression of the PB1 domain alone led to the formation of large mCh$^+$/GFP$^+$ structures, presumably oligomerized reporters, as well as the formation of some mCh$^+$/GFP$^-$ punctae. When expressed in HeLa mito-mCh-GFP cells, one or two copies of the LIR peptide fused to VHH$_{GFP}$ were not sufficient to induce lysosomal delivery of mito-mCh-GFP. Three repeats of the LIR peptide already significantly increased the formation of mCh$^+$/GFP$^-$ punctae and reduced protein levels of mitochondrial VDAC1 and COX4. As for the cytosolic target, expression of VHH$_{GFP}$-PB1-LIR was as efficacious in inducing mitophagy as VHH$_{GFP}$ fused to full length p62. Expression of the PB1 domain alone led to collapse of the mitochondrial network and formation of mCh$^+$/GFP$^-$ punctae, however to a lesser extent than PB1-LIR. Expression of the constructs comprised of concatenated LIR peptides as well as the PB1-LIR fusion in HeLa mito-mCh-GFP p62 K.O. cells led to results comparable to those observed in parental HeLa mito-mCh-GFP cells (Supplementary Fig. S6E–I), highlighting that co-recruitment of endogenous p62 is not required but rather the high local concentration of LIR peptides on the surface of mitochondria drives mitophagy. Accordingly, expression of VHH$_{GFP}$-PB1, which by itself is unable to interact with ATG8-like proteins, did not lead to formation of mCh$^+$/GFP$^-$ punctae in the absence of endogenous p62. Together, these data highlight the local concentration of LIR sequences as a key determinant for efficient autophagy-mediated targeted degradation and further establish the PB1-LIR construct as the minimal degron required for the lysosomal clearance of both cytosolic targets and mitochondria.

### Development of a p62 biodegrader for autophagy targeted degradation

The experiments in this study relied so far on overexpressed autophagy effectors. To prototype autophagy targeted degradation with an endogenous effector, we decided to generate a single-domain antibody against p62 to develop a bifunctional biodegrader. These relatively small antigen-binding fragments (~13 kDa) bind their target with high affinity and are increasingly receiving attention for their use as biodegraders[28,44–46]. To identify anti-p62 VHHs, two llamas were immunized with recombinant p62 lacking essential parts of the PB1 domain (His-MBP-p62$_{85-440}$), and VHH phage display libraries were generated and enriched by panning. Single VHH clones were triaged using a p62 enzyme-linked immunosorbent assay (ELISA) and cellular

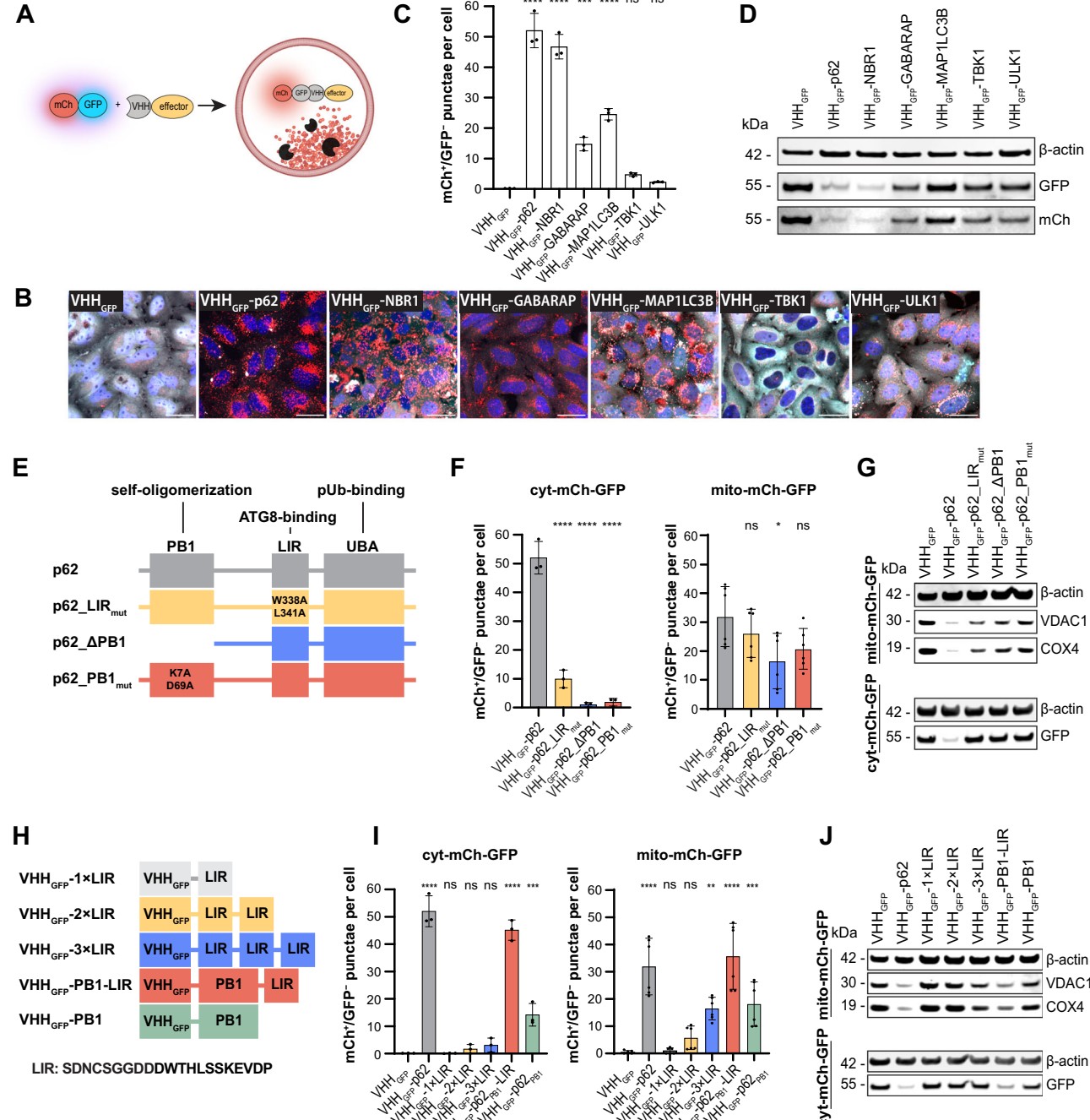

**Fig. 3 | Mutational studies reveal avidity as a requirement for autophagy targeted degradation. A** Schematic representation of the approach used to induce VHH$_{GFP}$-dependent proximity between the cytosolic reporter cyt-mCh-GFP and autophagy effectors. **B** Representative confocal fluorescence microscopy images showing stable HeLa cyt-mCh-GFP cells with inducible expression of fusions of VHH$_{GFP}$ and autophagy effectors for 72 h. Red = mCh, cyan = GFP, blue = nuclei. Scale bars are 30 μm. **C** Quantification of panel (**B**). **D** Immunoblotting of stable HeLa cyt-mCh-GFP cells with inducible expression of fusions of VHH$_{GFP}$ and autophagy effectors for 72 h. **E** Schematic representation of the p62 domain architecture and the tested mutants. **F** Quantification of mCh$^+$/GFP$^-$ punctae in stable HeLa mito-mCh-GFP and HeLa cyt-mCh-GFP cells with inducible expression of fusions of VHH$_{GFP}$ and mutated versions of p62 for 72 h. **G** Immunoblotting of stable HeLa mito-mCh-GFP cells and HeLa cyt-mCh-GFP cells with inducible

expression of fusions of VHH$_{GFP}$ and p62 mutants for 72 h. **H** Schematic representation of VHH$_{GFP}$-LIR fusion constructs tested for lysosomal translocation and the degradation of mitochondria and a soluble target. **I** Quantification of mCh$^+$/GFP$^-$ punctae in stable HeLa mito-mCh-GFP and HeLa cyt-mCh-GFP cells with inducible expression of LIR and PB1 containing constructs for 72 h. **J** Immunoblotting of stable HeLa mito-mCh-GFP and HeLa cyt-mCh-GFP cells with inducible expression of LIR and PB1 containing constructs for 72 h. Data are shown as mean and standard deviation from $n = 3$ (cyt-mCh-GFP) or $n = 6$ (mito-mCh-GFP) independent biological replicates. Statistical analysis was performed using an ordinary one-way ANOVA with multiple comparison of each data point against VHH$_{GFP}$-p62 (panel **F**) or VHH$_{GFP}$ (panels **C** and **I**) using Dunnett's multiple comparisons test. P-value summary: ns = ($P > 0.05$); *($P \le 0.05$); **($P \le 0.01$); ***($P \le 0.001$); ****($P \le 0.0001$). Exact p-values are shown in the source data.

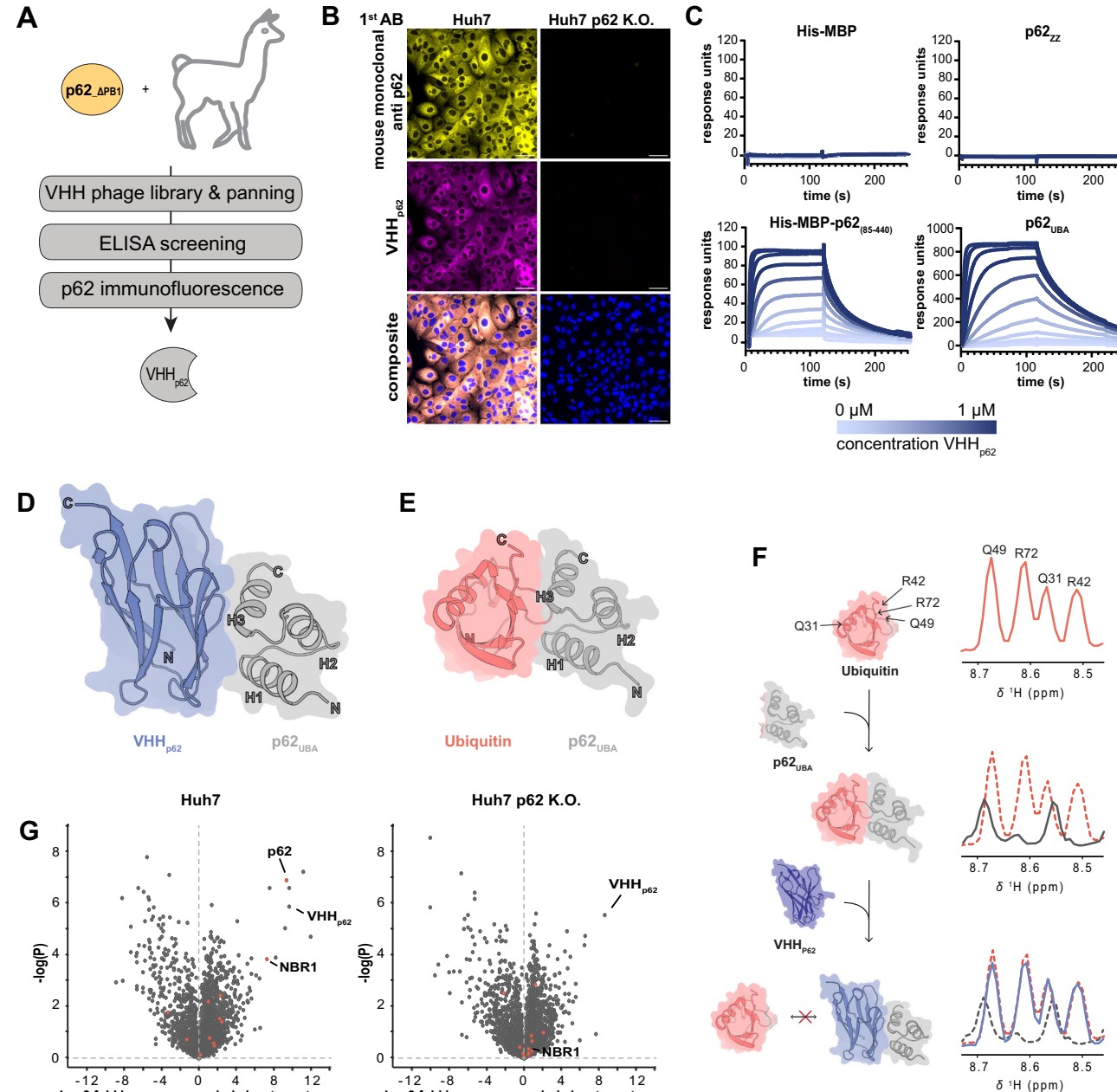

**Fig. 4 | Generation of a p62-binding VHH. A** Flow chart outlining the key steps in the generation and selection of VHH_p62. **B** Representative confocal fluorescence microscopy images showing parental Huh7 and Huh7 p62 K.O. cells. Cells were immunostained using an anti-p62 antibody and a periplasmic extract of *E. coli* expressing VHH_p62. Yellow = anti-p62 antibody, magenta = anti-VHH antibody, blue = nuclei. Scale bars are 30 μm. **C** SPR sensograms of VHH_p62 binding to various recombinantly produced truncation constructs of p62. Dissociation constant ($K_D$) values of 11 nM and 36 nM were calculated for His-MBP-p62_{85-440} and p62_{UBA}, respectively. **D** X-ray crystal structure of the p62_{UBA}:VHH_p62 complex. **E** Alphafold-Multimer prediction of the p62_{UBA}:ubiquitin complex illustrating a similar binding mode for ubiquitin as for VHH_p62 seen in panel (**D**). **F** 1D projections of the section in the 2D $^1$H,$^{15}$N-HSQC spectra of ubiquitin containing the resonances of residues Q49, R72, Q31, and R42. $^{13}$C,$^{15}$N-Ubiquitin was mixed with p62_{UBA} in a 1:1 ratio as well as with p62_{UBA} and VHH_p62 in a 1:1:1.2 ratio. Red: $^{13}$C,$^{15}$N-Ubiquitin alone. Black: $^{13}$C,$^{15}$N-Ubiquitin alone and p62_{UBA}. Blue: $^{13}$C,$^{15}$N-Ubiquitin, p62_{UBA} and VHH_p62. **G** Affinity proteomics to identify VHH_p62 binding partners in Huh7 and Huh7 p62 K.O. cells. Proteins that contain a UBA-domain are shown in red.

immunofluorescence (Fig. 4A and Supplementary Fig. S7A). This approach identified VHH 6C10 (hereafter called VHH_p62) as a highly selective binder of p62 (Fig. 4B and Supplementary Fig. S7B, C). Surface plasmon resonance (SPR) (Fig. 4C) revealed that VHH_p62 binds the His-MBP-p62_{85-440} immunogen and the p62 ubiquitin associated domain (p62_{UBA}) with an affinity of 11 nM and 36 nM, respectively, but not the His-MBP tag or the p62 ZZ domain (p62_{ZZ}). This data suggests that VHH_p62 is selective for the UBA domain of p62.

An X-ray co-crystal structure of VHH_p62 in complex with the p62 UBA domain was determined to 2 Å resolution to gain structural

insights into the binding of VHH_p62 to p62 (Fig. 4D). This structure reveals that VHH_p62 has a binding mode similar to ubiquitin as seen in an AlphaFold2-predicted[47] structure of a complex between p62_{UBA} and ubiquitin (Fig. 4E) as well as X-ray crystal structures of ubiquitin bound to UBA domains of other proteins[48]. To investigate if VHH_p62 binds the UBA domain in a ubiquitin-competitive manner, we employed nuclear magnetic resonance (NMR) spectroscopy using $^{13}$C and $^{15}$N labeled ubiquitin ($^{13}$C,$^{15}$N-Ub) (Fig. 4F and Supplementary Fig. S7D). Upon addition of p62 UBA, the peaks corresponding to residues Q49, R72, Q31, and R42 of $^{13}$C,$^{15}$N-Ub experienced a shift and broadening in the

[15]N-HSQC NMR spectrum, indicating interaction between the two proteins. Addition of VHH$_{p62}$ resulted in a spectrum of [13]C,[15]N-Ub that resembles the unbound state. These results indicate that VHH$_{p62}$ outcompetes ubiquitin for binding of the p62 UBA domain due to a much higher affinity. Since the UBA domain is a motif that occurs in several ubiquitin-binding proteins, we tested whether VHH$_{p62}$ binding is selective for p62. We used VHH$_{p62}$ as affinity reagent and performed pull down studies in parental and p62 knockout Huh7 cells (Fig. 4G and Supplementary Dataset 2). Global proteomics of immunoprecipitated samples from parental Huh7 cells identified p62 and several known interaction partners such as KEAP1, NBR1, NIPS1 and NIPS2. In contrast, neither p62 nor its interactors or other UBA-containing proteins were robustly enriched in p62 knockout cells, indicating that VHH$_{p62}$ is highly selective for the UBA domain of p62.

We next tested if VHH$_{p62}$ can be expressed in cells and developed into a biodegrader. For this purpose, we designed a bifunctional construct by fusing VHH$_{p62}$ to VHH$_{GFP}$ (VHH$_{GFP}$-VHH$_{p62}$) (Fig. 5A). Expression of VHH$_{GFP}$-VHH$_{p62}$ for 15 h in HeLa mito-mCh-GFP cells resulted in robust co-localization of the biodegrader as well as p62 with mitochondrial-tethered GFP (Fig. 5B). In contrast, the expression of VHH$_{GFP}$ alone resulted in VHH co-localizing with GFP while p62 remained cytosolic. This observation suggests that VHH$_{GFP}$-VHH$_{p62}$ can act as a heterobifunctional recruiter inducing proximity between GFP-tagged mitochondria and endogenous p62. Recruitment of endogenous p62 to mito-mCh-GFP resulted in the formation of mCh[+]/GFP[−] punctae and reduced levels of COX4 and VDAC1 in a pooled population as well as three individual clones (Fig. 5C−F and Supplementary Fig. S10A). Treatment with Bafilomycin A1 and PIK-III as well as knock out of p62 and ATG5 inhibited formation of mCh[+]/GFP[−] punctae and degradation of mitochondrial proteins (Supplementary Fig. S10B−D). Analysis of the individual clones shows slight variation in the efficiency of lysosomal delivery with clone 3 showing similar levels of mCh[+]/GFP[−] punctae as cells overexpressing the VHH$_{GFP}$-p62 fusion construct. To exclude that mitophagy induction is limited to HeLa cells, we tested VHH$_{GFP}$-VHH$_{p62}$ in human induced pluripotent stem cell (iPSC) derived NGN2 neurons expressing the mito-mCh-GFP reporter system and observed efficient formation of mCh[+]/GFP[−] punctae (Supplementary Fig. S10E). These data suggest that recruitment of endogenous p62 to the OMM is sufficient to induce mitophagy.

We then evaluated VHH$_{GFP}$-VHH$_{p62}$ for its ability to degrade soluble cyt-mCh-GFP (Supplementary Fig. S8A−E). Expression of VHH$_{GFP}$-VHH$_{p62}$ led to negligible formation of mCh[+]/GFP[−] punctae and no reduction of protein levels, despite strong VHH$_{GFP}$-VHH$_{p62}$ expression in the pooled population and in three individual clones. We hypothesized that VHH$_{p62}$ may interfere with p62 oligomerization, thereby impairing the avidity-dependent clearance of the cytosolic reporter. To explore this, we monitored p62 morphology in parental HeLa cells treated with Bafilomycin A1 and observed significantly decreased punctae formation upon expression of VHH$_{GFP}$-VHH$_{p62}$ both with and without treatment with Bafilomycin A1 (Supplementary Fig. S9A−C). To more directly evaluate if p62 oligomerization is inhibited by VHH$_{p62}$, we incubated recombinant full length p62 with various concentrations of recombinant VHH$_{p62}$ and performed size exclusion chromatography (SEC)[39]. In the absence of VHH$_{p62}$, p62 eluted in the void volume of the SEC column, indicating that it formed high molecular weight oligomeric structures. Addition of VHH$_{p62}$ resulted in a significant shift of p62 to an elution volume corresponding to a lower molecular weight range, similar to p62_PB1$_{mut}$ (Supplementary Fig. S9D). Furthermore, in a negative staining transmission electron microscopy (TEM) analysis of p62 oligomers, the addition of VHH$_{p62}$ significantly reduced the formation of high molecular weight oligomeric particles (Supplementary Fig. S9E, F).

Next, we evaluated whether multimeric or oligomeric cytosolic mCh-GFP would provide enough avidity for VHH$_{GFP}$-VHH$_{p62}$ to initiate degradation of a cytosolic target. We therefore generated a HaloTag-mCh-GFP-SNAPtag reporter, where multimerization can be induced upon addition of the chemical dimerizer HaXS8 (Supplementary Fig. S8F), as well as mCh-GFP reporter fused to the N-terminal domain of coilin (hereafter called coilin$_{NTD}$), which forms cytosolic oligomeric structures (Supplementary Fig. S8K)[49]. When expressed in HeLa HaloTag-mCh-GFP-SNAPtag cells, VHH$_{GFP}$-VHH$_{p62}$ did not initiate formation of mCh[+]/GFP[−] punctae, regardless of treatment with HaXS8, whereas expression of VHH$_{GFP}$-p62 led to robust lysosomal translocation as well as degradation of the target (Supplementary Fig. S8G−J). When expressed in HeLa coilin$_{NTD}$-mCh-GFP cells, however, VHH$_{GFP}$-VHH$_{p62}$ led to strong formation of mCh[+]/GFP[−] punctae and degradation of the target (Supplementary Fig. S8L−O). Together, these data underscore the importance of avidity, provided by high local concentration of the target, for successful degradation by VHH$_{GFP}$-VHH$_{p62}$.

## PINK1-VHH$_{p62}$ chimeras induce selective clearance of damaged mitochondria

The VHH$_{GFP}$-VHH$_{p62}$ biodegrader is a valuable tool to demonstrate that recruitment of endogenous p62 to mitochondria is sufficient to induce mitophagy, however, it has the limitations that it relies on an artificial interaction with GFP and targets all mitochondria. The selective removal of damaged mitochondria has the potential to enhance cellular health by restoring energy production, reducing oxidative stress, and preventing inflammation thus offering therapeutic potential to mitigate progression in various diseases[50–52]. In order to develop a selective biodegrader for damaged mitochondria, we fused VHH$_{p62}$ to PINK1. PINK1 is a key component of the mitochondrial quality control machinery that under physiological conditions gets continuously imported into mitochondria and degraded. Upon disruption of the mitochondrial membrane potential, PINK1 accumulates on the OMM where it activates Parkin to initiate mitophagy. We hypothesized that by fusing PINK1 with VHH$_{p62}$, p62 could be selectively recruited to the surface of damaged mitochondria. We therefore generated HeLa mito-mCh-GFP cells constitutively expressing PINK1-HA-VHH$_{p62}$ and PINK1-HA (Fig. 5G). We induced mitochondrial depolarization by treatment with the ionophore Valinomycin and observed accumulation of HA-tag, colocalization of p62 with mitochondria, as well as mCh[+]/GFP[−] punctae formation (Fig. 5H−J and Supplementary Fig. S10F, G). Under basal conditions, the construct is hardly detectable due to the fast turnover of PINK1[53]. When mitochondria were depolarized, both PINK1-HA and PINK1-HA-VHH$_{p62}$ got stabilized and colocalized with mitochondria. Moreover, in cells expressing PINK1-HA-VHH$_{p62}$, p62 colocalized with mitochondria, indicating its recruitment to impaired mitochondria via the PINK1-VHH$_{p62}$ chimera. Treatment with Valinomycin triggered fragmentation of mitochondria in cells expressing PINK1-HA but no mCh[+]/GFP[−] punctae were detectable, whereas cells expressing PINK1-HA-VHH$_{p62}$ showed strong formation of mCh[+]/GFP[−] punctae. These findings highlight that PINK1-VHH$_{p62}$ chimeras effectively enable the selective induction of mitophagy in response to mitochondrial damage, offering a targeted approach for mitochondrial quality control.

## Discussion

In this study, we developed a platform to identify and evaluate autophagy effectors that are capable of degrading mitochondria upon induced proximity. The first strategy is based on the intracellular expression of VHH fusion constructs and was leveraged to identify effectors that trigger mitophagy. The second strategy is based on a small-molecule approach using the self-labeling proteins HaloTag and SNAPtag. The latter approach allows more precise temporal control over the initiation of induced proximity and offers the possibility to use different concentrations of heterobifunctional dimerizers to fine-tune target occupancy. Using these two strategies, the autophagy cargo receptors p62 and NBR1, the ATG8-like proteins MAP1LC3 and

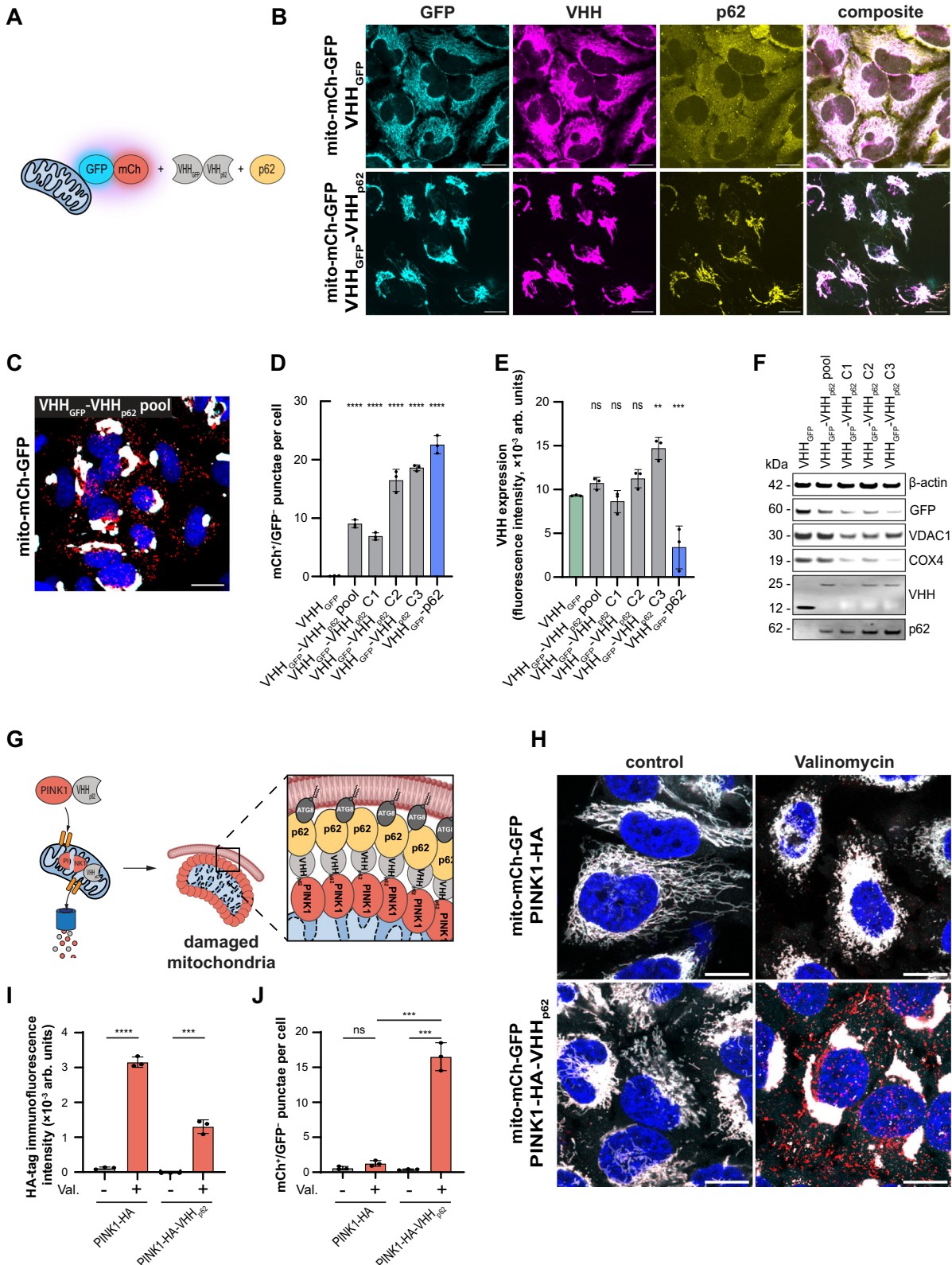

GABARAP, and the kinases ULK1 and TBK1 could be validated as effectors to clear mitochondria in a proximity-dependent manner. These findings align with a recent concurrent study, which reported mitochondrial degradation following the recruitment of ATG8-family proteins, ULK1, and ATG16L1[30], further demonstrating the robustness of this approach. We did not identify E3 ligases that triggered proximity-dependent mitophagy even though our focused cDNA library contained several candidates known to catalyze Lys63-dependent ubiquitination. In fact, many of the tested E3 ligases reduced levels of the mito-mCh-GFP reporter independently of lysosomal delivery (Supplementary Dataset 1). E3 ligase candidates inducing mitophagy may have been missed by our study because of the limited size of our focused library and the absence of mitochondrial damage to activate the PINK1-Parkin pathway[54]. In addition, steric

**Fig. 5 | Degradation of mitochondria using a p62 biodegrader. A** Schematic representation of the approach used to induce proximity between GFP-tagged mitochondria and p62 using VHH$_{GFP}$-VHH$_{p62}$. **B** Representative immunofluorescence imaging of HeLa mito-mCh-GFP cells with inducible expression of either VHH$_{GFP}$ or VHH$_{GFP}$-VHH$_{p62}$ for 15 h. Cyan = GFP, yellow = p62, magenta = VHH. Scale bars are 30 μm. **C** Representative confocal fluorescence microscopy image showing stable HeLa mito-mCh-GFP cells with inducible expression of VHH$_{GFP}$-VHH$_{p62}$ for 72 h. Red = mCh, cyan = GFP, blue = nuclei. Scale bars are 30 μm. **D** Quantification of mCh$^+$/GFP$^-$ punctae in stable HeLa mito-mCh-GFP cells with inducible expression of VHH$_{GFP}$, a pooled population as well as 3 selected clones expressing VHH$_{GFP}$-VHH$_{p62}$, or VHH$_{GFP}$-p62 for 72 h. **E** Quantification of VHH expression levels of panel (**B**). Fluorescence intensities were assessed by immunofluorescence microscopy. **F** Immunoblot of HeLa mito-mCh-GFP cells with inducible expression of VHH$_{GFP}$, a pooled population as well as 3 selected clones expressing VHH$_{GFP}$-VHH$_{p62}$, or VHH$_{GFP}$-p62 for 72 h. **G** Schematic representation of the approach used to induce proximity between damaged mitochondria and p62 using a PINK1-VHH$_{p62}$ chimera. **H** Representative confocal fluorescence microscopy images showing stable HeLa mito-mCh-GFP cells with constitutive expression of PINK1-HA and PINK1-HA-VHH$_{p62}$ for 72 h with and without induction of mitochondrial damage by Valinomycin treatment (1 μM, 4 h). Red = mCh, cyan = GFP, blue = nuclei. Scale bars are 30 μm. **I** Quantification of expression levels of HA-tag in the cells displayed in panel (**H**). HA-tag expression levels were assessed by immunofluorescence microscopy. **J** Quantification of mCh$^+$/GFP$^-$ punctae in the cells displayed in panel (**H**). Data are shown as mean and standard deviation from $n = 3$ independent biological replicates. Statistical analysis was performed using an ordinary one-way ANOVA with multiple comparison of each data point against VHH$_{GFP}$ using Dunnett's multiple comparisons test (panels **D** and **E**) or using an unpaired, two-sided $t$-test performed on untreated versus treated samples (panels **I** and **J**). $P$-value summary: ns = ($P > 0.05$); *($P \leq 0.05$); **($P \leq 0.01$); ***($P \leq 0.001$); ****($P \leq 0.0001$). Exact $p$-values are shown in the source data.

hindrance may have affected the activity of ligases that are sensitive to their orientation towards the target, which can be influenced by linker length or whether they are expressed as N- or C-terminal fusions[29]. Genome-wide cDNA library screens performed under different conditions will be needed to assess the global landscape of proximity-dependent mitophagy effectors in an unbiased and disease-relevant manner.

While p62, NBR1, GABARAP, MAP1LC3B, ULK1 and TBK1 were all able to induce proximity-dependent mitophagy, only p62 and NBR1 retained full efficacy against a cytosolic target. The ability of the ATG8-like proteins GABARAP and MAP1LC3B to deliver the cytosolic target to lysosomes was reduced, whereas the kinases ULK1 and TBK1 did not induce any degradation. Mechanistic studies using several mutants and truncated versions of the cargo receptor p62 revealed that a key requirement for degradation of soluble targets is the ability of the effectors to oligomerize. These findings are in line with reports showing that the affinities of various essential protein-protein interactions in the autophagy pathway are rather low[55,56] and that avidity, driven by oligomerization of cargo receptors, is required for interactions of p62 with ubiquinated cargo as well as membrane-bound ATG8-like proteins[39]. We suggest that the oligomerization of effectors may serve as an additional amplification step, allowing to exceed the signal activation threshold required to induce selective autophagy, which is reminiscent of the formation of polyubiquitin chains by E3 ligases on substrates targeted for proteasomal degradation.

Building on the findings that the autophagy cargo receptor p62 stood out as the strongest effector to induce proximity-dependent degradation of soluble as well as membrane-bound targets, we developed a VHH$_{p62}$-based biodegrader that recruits endogenous p62. Previous developments of p62-engaging small-molecule binders focused on binding the ZZ-domain and accelerate self-oligomerization[21]. The ability of p62 to form oligomers and undergo phase separation upon interaction with polyubiquitin chains is essential for lysosomal delivery of cargo because these processes increase avidity which stabilizes the interactions with autophagosome membranes[21]. The VHH$_{p62}$ described in this study, however, interacts with the UBA domain of p62 and disrupts its self-oligomerization. While the exact mechanism behind this disruption remains unclear, this feature allowed us to develop a biodegrader that selectively acts on scaffolded targets. The approach is based on the principle that avidity can be achieved either through self-oligomerization or through concentrated accumulation of effectors on the target. The data obtained from the experiments using p62 mutants suggest that scaffolded targets such as mitochondria are less dependent on the self-oligomerization of p62 to provide high local concentrations of LIR sites, because the templating effect can compensate a lack of self-oligomerization from the cargo receptor. Cyotosolic monomeric targets lack this templating effect and therefore can only become susceptible to degradation via autophagy when the effectors, self-oligomerize and provide the required high concentrations of LIR sites. Since a minimal construct of the p62 PB1 domain fused to an extended LIR domain is as efficacious as full-length p62, we propose that the local accumulation of LIR motifs and the concomitant phagophore recruitment is the driving force behind avidity-induced autophagy targeted degradation. Consequently, the biodegrader that inhibits p62 oligomerization proved ineffective in degrading the monomeric or multimeric cytosolic mCh-GFP model system. However, when the mCh-GFP tag was fused to the N-terminal domain of coilin, which is capable of forming oligomers, the local accumulation of p62 and LIR sites became sufficiently high to facilitate lysosomal recruitment and target degradation.

These results align with the observation that degraders engaging kinases or the ATG8 family were mainly described for the degradation of scaffolded targets such as mitochondria[23] and protein aggregates[22]. In contrast, p62-engaging degraders have been also reported for the clearance of soluble targets such as methionine aminopeptidase 2 and the androgen receptor[21]. This underscores the importance of identifying the optimal effector-target combination and highlights the potential to develop degraders that are selective for the aggregation state of the target protein. Such selectivity might be particularly relevant for the development of therapeutics against aggregation-prone proteins like SOD1 or alpha-synuclein, which serve essential functions in their non-aggregated states but are pathogenic when aggregated[57]. The same principle could be extended to design degraders targeting whole organelles, ensuring they selectively act on damaged or dysfunctional organelles expressing specific protein markers that distinguish them from healthy ones. Aggregation and avidity-induced degradation technologies via TRIM21-mediated ubiquitination have recently been reported and show potential for clearance of pathogenic protein aggregates[58–60]. Autophagy targeted degradation is a promising alternative avenue to induce the selective clearance for aggregated disease drivers or even whole organelles.

We established the proof of concept for avidity-induced targeted degradation mediated by the p62 biodegrader, using mitochondria as a scaffolded target by engaging GFP on their surface. However, these experiments did not distinguish between healthy and damaged mitochondria, a critical consideration for the development of treatments for diseases linked to mitochondrial dysfunction. To address this limitation, we fused VHH$_{p62}$ to PINK1, demonstrating that avidity-induced degradation via p62 can selectively target damaged mitochondria thus offering a more precise approach for targeted mitochondrial quality control. We hypothesize that this degradation paradigm only works because PINK1-VHH$_{p62}$ is not degraded via autophagy while in its cytosolic state, allowing its transport to mitochondria and fast turnover under physiological conditions. Accumulation of the chimera on the OMM of depolarized mitochondria increases the local concentration of VHH$_{p62}$ and provides avidity necessary for lysosomal delivery. Further studies exploring the

functional consequences of selective clearance of dysfunctional mitochondria by targeted degradation will be critical to determine the feasibility of this approach for the development of novel therapies. In conclusion, targeted mitophagy via avidity-induced autophagic degradation could pave the way for new treatments in diseases associated with mitochondrial dysfunction.

## Methods

### Reagents
Bafilomycin A1 (19-148, Merck) was dissolved at 100 μM in DMSO. SNAP-Cell® 647-SiR (S9102S, New England Biolabs) was dissolved at 1 mM in DMSO. Janelia Fluor®-646 HaloTag® (GA112A, Promega) was dissolved at 1 mM in DMSO. Doxycyclin-Hyclat (D5207, Merck) was dissolved at 1 mg mL$^{-1}$ in deionized water. HaXS8 (4991, Tocris bioscience), Valinomycin (V3639, Merck), and PIK-III (Novartis) were dissolved at 10 mM in DMSO. Unless prepared freshly, stock solutions were stored as frozen aliquots at −20 °C.

### Antibodies
The anti-APG5L/ATG5 (ab228668, 1:500 for WB), anti-COX4 (ab14744, 1:400 for WB), anti-mCherry (ab125096, 1:500 for WB), anti-TOMM20 (ab283317, 1:500 for WB) and anti-LAMP1 (ab25630-100, 1:200 for IF) antibodies were purchased from Abcam. The anti-p62 (610833, 1:500 for WB and IF) antibody was purchased from BD Biosciences. The anti-β-actin (3700S, 1:5000 for WB), anti-COX4 (4850S, 1:800 for WB), anti-GFP (2956S, 1:500 for WB), anti-HA (3724S, 1:500 for IF and WB) and anti-VDAC1 (4866S, 1:500 for WB) antibodies were purchased from Cell Signaling Technology. The anti-VHH-ifluor647 conjugate (A02170-200, 1:500 for IF) antibody was purchased from Genescript. The anti-VHH (200-401-GM6, 1:500 for WB) antibody was purchased from Rockland. The anti-HaloTag antibody, (G921A, 1:500 for WB) was purchased from Promega. The Alexa Fluor™ 488 Goat anti-Mouse IgG (A11029, 1:1000 for IF), Alexa Fluor™ 647 Donkey anti-Mouse IgG (A31571, 1:1000 for IF), Alexa Fluor™ 647 Donkey anti-Rabbit IgG (A31573, 1:1000 for IF), Alexa Fluor™ 488 Goat anti-Rabbit IgG (A11008, 1:1000 for IF) and Alexa Fluor™ 405 Goat anti-Mouse IgG (A31553, 1:500 for IF) antibodies were purchased from Invitrogen. The IRDye® 800CW Donkey anti-Rabbit IgG (926-32213, 1:5000 for WB) and IRDye® 680RD Donkey anti-Mouse IgG (926-68072, 1:5000 for WB) antibodies were purchased from LICORbio.

### Cell culture and development of stable cell lines
HeLa and Huh7 cells (ATCC) were cultured in DMEM (Gibco, Thermo Fisher Scientific) supplemented with 10% fetal bovine serum and 1% Antibiotic-Antimycotic (Gibco, Thermo Fisher Scientific). Cells were cultured at 37 °C in a humified chamber under an atmosphere supplemented with 5% CO$_2$.

Hela cell lines expressing mito-mCh-GFP, mito-mCh-GFP-SNAP-tag, cyt-mCh-GFP, HaloTag-mCh-GFP-SNAPtag, coilin$_{NTD}$-mCh-GFP, PINK1-HA, and PINK1-HA-VHH$_{p62}$ were generated by lentiviral transduction followed by selection using blasticidine (4 μg mL$^{-1}$; Gibco, Themofisher Scientific) or puromycin (2 μg mL$^{-1}$; Gibco, Themofisher Scientific) for 3 passages. Where necessary, single clones were obtained by limiting dilution in 96-well plates. Wells containing single colonies were characterized using fluorescence microscopy and clones with minimal basal formation of mCh$^+$/GFP$^-$ punctae were used for further experiments.

To generate the doxycycline-inducible cell lines, HeLa mito-mCh-GFP, HeLa mito-mCh-GFP-SNAPtag, HeLa cyt-mCh-GFP, HeLa HaloTag-mCh-GFP-SNAPtag, or HeLa coilin$_{NTD}$-mCh-GFP cells were cultured in 96 well plates (Techno Plastic Products AG, 92448) to 90% confluency and were co-transfected with plasmids encoding the respective VHH$_{GFP}$-autophagy effector fusion and transposase[61] (100 ng each) using Lipofectamine 3000 Transfection Reagent (Invitrogen, Thermo Fisher Scientific) for 48 h, according to the manufacturer's

instructions. Selection for 2 weeks was performed using puromycin (2 μg mL$^{-1}$; Gibco, Themofisher Scientific). Cells were passaged at least 3 times before further experiments.

P62 knockout cells were generated through electroporation of Cas9-sgRNA ribonucleoprotein complex using Neon™ Transfection System (Invitrogen). Two Alt-R™ CRISPR-Cas9 crRNA (IDT) targeting the SQSTM1 gene (GGCGCCTCCTGAGCACACGG and GCAGCCATCG-CAGATCACAT) were annealed to Alt-R® CRISPR-Cas9 tracrRNA (IDT, N1072532) to obtain 50 μM gRNA. 1 μL of each gRNA was mixed with 10 μg Alt-R™ S.p. Cas9 Nuclease V3 (IDT, 1081058) and electroporated into 500,000 cells. Single clones were obtained by limiting dilution in 96-well plates. To confirm knockout of the SQSTM1 gene, genomic DNA from single clones was extracted using DNeasy Blood & Tissue Kit (Qiagen, 69504) and the SQSTM1 gene was amplified by PCR and and sequenced. The sequencing primers for SQSTM1 were AGCCCCA-CAGTGACGACAGA and AAGGTTCCTGCCACCCTGAG. Absence of p62 was further validated by immunoblot analysis.

ATG5 knockout cells were generated through co-transfection of equimolar ratios (20 fmol) of pYJA5 ATG5 4x sgRNA[62] and pCAG Cas9, using Lipofectamine 3000 (Invitrogen, Thermo Fisher Scientific) according to the manufacturer's instructions. Culture media change of transiently transfected cells was performed the following day. Cultures were further expanded for 2 weeks. Confluent cell cultures were dissociated for 15 min at 25 °C, using TrypLE Express solution (Gibco, Thermo Fisher Scientific), and further quenched in equal volumes of culture media. Single cell suspensions were further resuspended in FACS-buffer (DPBS, Gibco, Thermo Fisher Scientific, 5 mM EDTA, Invitrogen, Thermo Fisher Scientific, 10% FBS). Cell suspensions were adjusted to 10$^6$ cells mL$^{-1}$, filtered through a 35 μm nylon mesh (Corning) and TagBFP$^+$ singlets were sorted into 5 mL 4 °C cooled culture media with the FACSAria Fusion (Beckton Dickinson) collecting minimum 10$^5$ cells per polyclonal cell line. Gating strategy was set in comparison to untransfected control cell lines. Polyclonal TagBFP$^+$ cultures were further expanded for 1 week in full growth medium, in order to reach 80–90% confluency of a 10 cm (TPP) dish. Single clones were further generated by limited dilution in 96 well plates and characterized by western blot.

NGN2 lines were generated following a previously described protocol[63]. Human induced pluripotent stem cells (hiPSCs) were derived from dermal fibroblasts isolated from a healthy 32-year-old female human donor (Invitrogen, C-013-5C) using CytoTune™-iPS 2.0 Sendai Reprogramming Kit (Invitrogen, A16517). The hiPSCs were cultured in flasks coated with Matrigel® Basement Membrane Matrix (Corning, 354248) in mTeSR1 (STEMCELL Technologies, 85857), supplemented with Penicillin-Streptomycin (Gibco, 15070-063), 500 μg mL$^{-1}$ Geneticin (Gibco, 10131035), and 10 μM ROCK inhibitor (Sigma-Aldrich, Y-27632). Stable cell lines expressing mito-mCh-GFP were then generated by lentiviral transduction followed by selection using blasticidine (4 μg mL$^{-1}$; Gibco, Themofisher Scientific) for 3 passages. For differentiation into NGN2 neurons, these cells were plated in in flasks coated with Matrigel® Basement Membrane Matrix (Corning) in DMEM/F-12, GlutaMAX™ supplement (Gibco, 10565-018) supplemented with 2% B-27™ (Gibco, 17504−044), Penicillin-Streptomycin (Gibco), 1% N-2 (Gibco, 17502048), 10 ng mL$^{-1}$ human EGF recombinant protein (Gibco, PHG0315), 10 ng mL$^{-1}$ human FGF-Basic full length recombinant protein (Gibco, CTP0263), 10 μM ROCK inhibitor (Sigma-Aldrich, Y-27632), and 1 μg mL$^{-1}$ doxycycline (Sigma-Aldrich, D5207) for 72 h. Progenitors were kept frozen in fetal bovine serum (Gibco, A5256701), supplemented with 5% DMSO. These cells were then seeded into 384-well plates (Greiner, 781096) coated with Matrigel® Basement Membrane Matrix (Corning) at a density of 40,000 cells per well in Neurobasal Medium (Gibco, 21103049) supplemented with Penicillin-Streptomycin (Gibco), 1% N-2 (Gibco), 2% B-27™ (Gibco), 10 ng mL$^{-1}$ recombinant human BDNF protein (R&D Systems, 248-BDB), and 10 ng mL$^{-1}$ recombinant human GDNF protein (R&D

Systems, 212-GD) for 48 h before transduction with lentiviruses allowing constitutive expression of VHH$_{GFP}$ or VHH$_{GFP}$-VHH$_{p62}$. Cells were cultured for additional 7 days before imaging with change of medium every 48 h.

## Molecular cloning

The base Gateway-compatible iVHH-DEST and iHaloTag-DEST plasmids were designed based on a previously described backbone[61] using the ePiggyBac transposable element. A Tet-On doxycycline inducible promoter, a puromycin resistance cassette, and a VHH or HaloTag cassette followed by (G$_4$S)$_2$ linker and a Gateway ready attR1-CmR-ccdB-attR2 cassette were used. Plasmids were constructed by GENE-WIZ (Azenta Life Sciences). Truncated mutant inserts were generated via synthesis and standard subcloning techniques and then shuttled into the Gateway pENTR221 vector backbone by Thermo Fisher Scientific Inc. (GeneArt; Carlsbad, CA). Full length inserts with targeted mutations were generated using standard site-directed mutagenesis techniques and cloned into the iVHH-DEST backbone using Gateway LR cloning by BioInnovatise (Rockville, MD). All remaining inserts (including truncation mutants) were cloned into the iVHH-DEST backbone or iHaloTag-DEST backbone using Gateway™ LR Clonase™ II Enzyme mix (Invitrogen, 11791020). Sequences are provided in Supplementary Dataset 3.

Lentiviral vectors used to generate cell lines expressing mito-mCh-GFP, mito-mCh-GFP-SNAPtag, cyt-mCh-GFP, HaloTag-mCh-GFP-SNAPtag, and coilin$_{NTD}$-mCh-GFP were generated using the CAG promoter, blasticidine resistance, and the previously described sequence for mCh-GFP-FIS1(101-152)[64], the mCh-GFP-FIS1(101-152) sequence linked to SNAPtag by a G4S spacer, mCh-GFP, HaloTag and SNAPtag linked to mCh-GFP by a 3× G4S spacer, or coilin(1-97) linked to mCh-GFP by a 3× G4S spacer, respectively. Lentiviral vectors used to generate cell lines expressing PINK1-HA and PINK1-HA-VHH$_{p62}$ were generated using the EF1α-promoter, puromycin resistance, and the PINK1 sequence linked to HA-tag or HA-tag, followed by three repeats of a G4S spacer and the sequence for VHH$_{p62}$. Plasmid synthesis and lentiviral packaging were performed by VectorBuilder (VB170815-1133qek, VB210615-1191kqe, and VB220517-1101xpp).

For mitochondrial degradation in Ngn2 neurons, lentiviral vectors, which allow constitutive expression of the respective proteins as N-terminal fusion with VHH$_{GFP}$ were constructed by VectorBuilder (VB230517-1309qzx, VB230517-1311wny, VB230517-1315cba, VB230517-1322axc, VB241129-1115erg, VB241129-1193hys, VB240605-1463ugk, and VB241217-1085bqe).

For mitochondrial degradation in Huh7 cells, lentiviral vectors, which allow constitutive expression of mito-mCh-GFP-SNAPtag and HaloTag-p62 or HaloTag-GABARAP were constructed by VectorBuilder (VB240113-1015nrn and VB240113-1016eme).

For recombinant expression of p62$^{WT}$ and p62$^{K7A/D69A}$ in *E. coli*, residues 2-440 of wildtype human full length p62$^{WT}$ (uniprot ID Q13501) and the double mutant p62$^{K7A/D69A}$ were ordered from GeneArt AG. The protein construct was encoded in a modified pACE-GP12 vector with an N-terminal combination of a Hexahistidine-tag (His$_6$), the IgG-binding domain of protein A (ZZ-tag)[65], a glycine-serine (lin[3xG$_3$S]) spacer and a HRV3C cleavage site (Px) as well as with a C-terminal Avi-tag.

## Fluorescence microscopy

Confocal fluorescence imaging was performed using a Yokogawa CV7000 microscope equipped with an automated stage incubator (37 °C, 5% CO$_2$), 405 nm, 488 nm, 561 nm, and 640 nm laser lines, 405/488/561 or 405/488/640 confocal paths, BP445/45, BP525/50, BP 617/67, and BP676/29 emission filters, and a 2560 px × 2160 px sCMOS camera (6.5 micron pixel size). Unless otherwise stated, a 20× UPLSAPO (0.75 N.A. air immersion), 40× UPLSAPO (0.95 N.A. air immersion), or 60× UPLSAPO (1.2 N.A. water immersion) objectives

were used. Unless otherwise stated, two technical replicates were acquired for each condition with 16 fields of view for each technical replicate.

High-resolution images were obtained using a ZEISS LSM900 microscope equipped with 405 nm, 488 nm, 561 nm, and 640 nm laser lines, a 63× Plan-Apochromat objective (1.4 N.A. oil immersion) and an Airyscan 2 detector unit. Images were processed using the ZEN Blue software.

## Fluorescence microscopy assay to detect mCh$^+$/GFP$^-$ punctae

Cells were plated in 96-well plates (Greiner, 655097) at a density of 12,000 cells per well and kept for 15 h at 37 °C before treatment with 1 µg mL$^{-1}$ doxycycline for 72 h. 2 µg mL$^{-1}$ Hoechst 33342 Trihydrochloride trihydrate (Invitrogen, H3570) was added for 1 h. The medium was changed, and the cells were imaged using a Yokogawa CV7000 confocal microscope at 37 °C with 5% CO$_2$ atmosphere. For testing autophagy-dependent formation of mCh$^+$/GFP$^-$, 100 nM Bafilomycin A1 or 5 µM PIK-III was added 15 h or 72 h prior to imaging, respectively. To assess PINK1-HA-VHH$_{p62}$-mediated mCh$^+$/GFP$^-$ formation, cells were treated with Valinomycin (1 µM) for 4 h before the medium was exchanged and cells were incubated in full growth medium at 37 °C for 72 h. For time-lapse microscopy experiments, cells were plated in 96-well plates (Greiner, 655097) at a density of 30,000 cells per well and kept for 15 h at 37 °C and 5% CO$_2$ before treatment with 2 µg mL$^{-1}$ Hoechst 333342 for 1 h. The medium was exchanged, the cells were treated with 1 µg mL$^{-1}$ doxycycline, and immediately subjected to imaging using a Yokogawa CV7000 confocal microscope at 37 °C with 5% CO$_2$ atmosphere. Pictures were taken every hour for 50 h.

## Immunofluorescence staining

Cells were plated in glass-bottom 96-well plates (IBIDI, 89626) at a density of 40,000 cells per well and kept for 15 h at 37 °C and 5% CO$_2$ before treatment with 1 µg mL$^{-1}$ doxycycline for the indicated times. The cells were then fixed with 4% paraformaldehyde (Electron Microscopy Sciences, 15714-S) at 25 °C for 15 min, washed with PBS and incubated with blocking buffer comprised of PBS, 0.1% Triton X-100 (Sigma-Aldrich, 93443), and 2% bovine serum albumin (Sigma-Aldrich, A9418) for 15 h at 4 °C. The cells were then incubated with primary antibodies in blocking buffer for 15 h at 4 °C, washed 3 times with PBS, incubated with secondary antibodies in blocking buffer for 4 h at 25 °C, washed 3 times with PBS, and imaged using a Yokogawa CV7000 confocal microscope or a LSM900 microscope with Airyscan2.

## Induced proximity with HaXS8

HeLa mito-mCh-GFP-SNAP cells were grown to 90% confluency and seeded into 6-well plates for immunoblotting assays or 384-well plates (Greiner, 781097) for imaging assays and cultured for 15 h at 37 °C and 5% CO$_2$ in growth medium supplemented with 1 µg mL$^{-1}$ doxycycline. The cells were then treated with HaXS8 at the indicated concentrations for 72 h in presence of 1 µg mL$^{-1}$ doxycycline. For fluorescence microscopy analysis, the cells washed with PBS and incubated with growth medium containing 2 µg mL$^{-1}$ Hoechst 33342 Trihydrochloride trihydrate for 1 h. The medium was then changed and the cells were imaged live using a Yokogawa CV7000 confocal microscope at 37 °C with 5% CO$_2$ atmosphere. The cells were then incubated in growth medium containing 200 nM Janelia Fluor®−646 HaloTag® or 200 nM SNAP-Cell® 647-SiR for 1 h. The medium was changed and the cells were fixed using 4% paraformaldehyde for 15 min at 25 °C. The cells were washed with PBS and imaged.

For HaXS8-induced mitophagy in Huh7 cells, cells were seeded in 384-well plates (Greiner, 781956) at a density of 3000 cells per well one day before lentiviral transduction. Briefly, lentiviruses encoding mito-mCh-GFP-SNAPtag and either HaloTag-GABARAP or HaloTag-p62 were added to cells in growth medium containing 8 µg mL$^{-1}$ of polybrene. Samples were then spinfected for 1 h at 800 × *g* at 37 °C and were

further incubated at 37 °C with 5% $CO_2$. The medium was changed 24 h post-transduction to remove residual lentivirus and the cells were further incubated overnight before being treated with 100 nM HaXS8 for 72 h. The growth medium was changed, the cells were incubated with 2 µg mL$^{-1}$ Hoechst 33342 for 45 min and the medium was changed for FluoroBrite DMEM (Gibco, Thermo Fisher Scientific) supplemented with 10% FBS and 1% Penicillin-Streptomycin. Samples were imaged using a Zeiss LSM980 confocal microscope at 37 °C and 5% $CO_2$ atmosphere.

## Image analysis

Fluorescence microscopy images were processed using ImageJ version 1.53t and analyzed using a custom pipeline in CellProfiler version 4.2.4. In brief, nuclei were identified as primary objects, and a cell mask was drawn based on combined GFP and mCh fluorescence signal using adaptive Otsu's thresholding. Punctae were identified by enhancing speckles in the channel depicting the mCh signal using the "EnhanceOrSuppressFeatures" module, followed by primary object identification using adaptive Otsu's thresholding. For each of these objects, the ratio of mCh over GFP signals was calculated. Objects with a mean mCh/GFP intensity lower than 1 were excluded from further analysis. For mitochondrial targets, an additional filtering step was performed, accounting for the form factor of the identified objects to exclude elongated, tubular structures. Of the remaining objects, only those within the boundaries of a mask depicting the area with detectable fluorescence intensity surrounding nuclei were used for further analysis. The number of detected objects was divided by the number of detected nuclei in each field of view. The mean value of the obtained measurement results was used for further analyses. Data were further processed using KNIME version 4.6.5 and plotted using GraphPad Prism version 10.1.2 or TIBCO Spotfire Analyst version 12.0.9.

## Statistical analyses

For all experiments, the number of replicates, error bars, and statistical significance are defined in the relevant figure legends. Statistical analysis was performed using GraphPad Prism version 10.1.2 for Windows. Unless otherwise stated, statistical significance was calculated by using one-way ANOVA followed by Dunnett's multiple comparisons test or paired, parametric t-test.

## Cell lysis and immunoblotting

Cells were washed with cold PBS, treated with cold RIPA lysis buffer (Thermo Scientific, 89901) supplemented with cOmplete ULTRA EDTA-free protease inhibitor cocktail (Roche, 5892791001) and sonicated with a tip-sonicator (15 pulses, 40% intensity). The protein concentration of each whole cell lysate was determined using a BCA protein quantification assay (Thermo Scientific, 23228). Lysates were then diluted in NuPAGE LDS Sample Buffer (Invitrogen NP0007) supplemented with NuPAGE Sample Reducing Agent (Invitrogen, NP0009), loaded into precast NuPAGE 4-12% Bis-Tris protein gels (Invitrogen, WG1403BOX), and separated by electrophoresis at 120 V for 2 h using NuPAGE MES SDS running buffer (Invitrogen, NP0002). The gels were transferred onto 0.2 µm PVDF membranes (Bio-Rad, 1704157) using the Trans-Blot Turbo Transfer System (Bio-Rad, 1704150EDU) for 7 min and then blocked for 30 min at 25 °C using Intercept blocking buffer (Li-COR Biotech, 927-70001). Membranes were incubated with appropriate primary antibodies in Intercept blocking buffer supplemented with 0.1% TWEEN 20 (Bio-Rad, 927-70001). Membranes were washed 3 times with PBS supplemented with 0.1% TWEEN 20 and incubated with secondary antibodies for 2 h at 25 °C. Membranes were washed 3 times with PBST and visualized using the ODYSSEY Infrared Imaging System.

## Llama immunization and generation of VHH$_{p62}$

Llama immunization campaign was performed by QVQ Holding B.V (Utrecht, Netherlands). In brief, recombinant His-MBP-p62(85-440) was used to immunize two llamas. Two immune VHH phage display libraries were prepared from mRNA isolated from peripheral blood lymphocytes taken from the llamas. Phages displaying VHHs were enriched for binding to His-MBP-p62(85-440)-biotin in two rounds of selection. Single clones were then picked, grown and periplasmic extracts were produced from each clone. Binding of the VHH candidates contained in the periplasmic extracts was determined by ELISA using His-MBP-p62(85-440)-coated and His-MBP-coated plates. To further characterize selected VHH candidates, Huh7 and Huh7 p62 K.O. cells were fixed with paraformaldehyde (4%) for 15 min at 25 °C, washed with PBS, permeabilized with 0.1% TritonX in PBS containing 2% BSA for 2 h at 25 °C and incubated with 1:500 anti-p62 (BD Bioscience, 610833) and 1:500 periplasmic extract in blocking buffer for 15 h at 4 °C. Cells were washed three times with PBS and incubated with 1:500 MonoRab™ Rabbit Anti-Humanized VHH Antibody for 15 h at 4 °C in blocking buffer. Cells were then washed three times with PBS and were incubated with 1:1000 Alexa Fluor™ 488 Goat anti-Rabbit, 1:1000 Alexa Fluor™ 647 Donkey anti-Mouse IgG and 4 µg mL$^{-1}$ Hoechst 33342 Trihydrochloride trihydrate in blocking buffer for 2 h at 25 °C. Cells were washed three times with PBS and imaged using an IN Cell analyzer 2200 (GE Healthcare Life Sciences).

## Surface plasmon resonance

SPR experiments were performed using a Biacore T-200 instrument (GE Healthcare) in 50 mM Tris, 150 mM NaCl, 1 mM TCEP, 0.05% Tween-20 at pH 7.4 and 25 °C. Biotinylated His-MBP-avi (His-MBP), His-MBP-p62(85-440)-Avi (His-MBP-p62$_{85-440}$), p62(387-446)-Avi (p62$_{UBA}$), or p62(122-181)-Avi (p62$_{ZZ}$) were immobilized on high-affinity streptavidin sensor chips (Cytiva) at densities of 520 RU, 700 RU, 500 RU, and 570 RU, respectively. VHH$_{p62}$ was serially diluted in the running buffer and injected individually. Cycles were run at a flow rate of 30 µL min$^{-1}$ with 120 s contact time and 300 s dissociation time with syringe wash steps between injections. Data analysis was performed using the Biacore Evaluation Software (GE Healthcare). All data were processed by double referencing to the reference channels and blank injection and the resulting sensorgrams were fitted using a kinetic model assuming a 1:1 binding mode for $K_D$ estimation.

## Immunoaffinity purification and proteomic analysis of VHH$_{p62}$ binding partners

One mg of cell lysates from parental Huh7 and Huh7 p62 K.O. cells ($n = 1$) were each incubated for 15 h at 4 °C under agitation with 2 µg of VHH$_{p62}$-His (6C10) and 50 µL of anti-His tag magnetic beads (MBL Life science, D291-11), pre-equilibrated in lysis buffer. Samples were also incubated with magnetic beads without VHH$_{p62}$ to determine non-specific binding to the magnetic matrix. Magnetic beads were then washed three times with lysis buffer. The beads were eluted twice with non-reducing sample buffer for 5 min at 37 °C on a shaker. After adding 0.1 M DTT, both eluates per sample were pooled and heated for 10 min at 60 °C on a shaker. Samples were separated on a SDS gel (Mini-Protean TGX precast gels, gradient 4–20%, Bio-Rad, 4561094) with a running distance of 3 cm. Gels were fixed (1% acetic acid, 30% ethanol, 20 min) and stained with Instant blue (Abcam, ab119211). After thorough washing with water, samples were in-gel digested with trypsin/lysC in 3 fractions per lane (<40 kDa, 40–150 kDa, >150 kDa). Extracted peptides were vacuum-dried, resuspended in water containing 0.1% formic acid, and analyzed in triplicates by trapping with a PepMap Neo C18 2 µm, 75 µm × 150 mm column (ThermoFisher) and separation on a 3rd generation Aurora column (Ion Optics, C18 1.7 µm, 250 mm × 75 µm) using a Vanquish Neo HPLC (ThermoFisher) and the following gradient of eluents A(0.1% formic acid in $H_2O$) and B (0.1% formic acid in 80% acetonitrile): 2 min 6% B, 60 min from 6% B to 31% B, 5 min from 31% B to 50% B, 5 min from 50% B to 95% B, 8 min 95% B and 8 min 6% B at a flow rate 250 nL min$^{-1}$. Eluting peptides were electrosprayed (1.6 kV spray voltage, 305 °C capillary temperature) in positive ion mode into

an ThermoFisher Orbitrap Eclipse Tribrid tandem mass spectrometer equipped with a Nanospray Ion Flex Source (ThermoFisher) and a column oven. Total acquisition time was set to 88 min. Orbitrap resolution was 120,000, scan range was 375–1200 m/z, RF Lens was 30%, AGC target standard, maximum injection time mode: auto, data type: profile. Filters: monoisotopic peak determination: peptide, intensity threshold $10^4$, charge states 2–6, dynamic exclusion duration: 60 s, mass tolerance width 20 ppm. Data dependent acquisition: cycle time, time between master scans: 3 s. ion trap MS/MS settings: HCD, collision energy mode/type/%: fixed/normalized/30. Scan rate: rapid, range mode: auto. Maximum injection time mode auto, data type: centroid. Raw files were analyzed with MaxQuant v2.2.0 and searched against the Human_UP000005640_reviewed UniProtKB database and the added sequence of $VHH_{p62}$ with standard parameters with run-to-run enabled. MaxQuant results were analyzed with Perseus v2.0.7.0 with standard parameters. Quality check was run for significantly enriched proteins by checking peptide availability.

## Size exclusion chromatography

Analytical Size Exclusion Chromatography (aSEC) experiments were carried out on an Agilent 1200 series Gradient High-Performance Liquid Chromatography (HPLC) system equipped with Agilent 1260 Infinity Diode Array Detectors DAD G1315C as well as an Agilent 1260 Infinity Fluorescence Detector (G1321B). All samples were injected with a volume of 10 µL onto Superdex 75 or Superdex 200 increase 5/150 GL columns (Cytiva) at a flow rate of 0.3 mL min$^{-1}$ and at 25 °C, using the Agilent 1260 Infinity Standard Autosampler (G1329B).

The impact of K7A/D69A mutations on p62 oligomerization was investigated by the injection of p62$^{WT}$ and p62-PB1$_{mut}$ at a concentration of 50 µM each. Interaction studies between p62$^{WT}$ and $VHH_{p62}$ were performed by mixing p62$^{WT}$ at a fixed concentration of 12.5 µM in 50 mM HEPES pH 7.0, 250 mM NaCl, 5% Glycerol and 1 mM TCEP with increasing $VHH_{p62}$ concentrations to achieve molar ratios of p62$^{WT}$:$VHH_{p62}$ of 1:0, 1:0.5, 1:1, 1:2, 1:4, and 1:8. The samples were incubated for 2 h at 4 °C prior to analysis. Binding between $VHH_{p62}$ and $p62_{UBA}$ was investigated by incubating $VHH_{p62}$ (20 µM) and $p62_{UBA}$ for 2 h at 4 °C before injection onto a Superdex 75 Increase 5/150 GL.

## Nano differential scanning fluorimetry

Nano differential scanning fluorimetry (nanoDSF) experiments were performed using a Prometheus Panta system together with Prometheus NT48 Series nanoDSF Grade Standard Capillaries. NanoDSF was used to determine the thermal stability of $VHH_{p62}$ or its mutants alone and in the presence of $p62_{UBA}$. For this, a mixture of $p62_{UBA}$ and $VHH_{p62}$ at concentrations of 20 µM and 10 µM, respectively, in buffer containing 50 mM HEPES pH 7.0, 250 mM NaCl and 1 mM TCEP was incubated for 1 h before measurement by nanoDSF. Samples were heated from 25 °C to 90 °C using a temperature ramp of 0.5 °C min$^{-1}$ and thermal denaturation was monitored by fluorescence detection at 330 and 350 nm. The melting temperature ($T_m$) was obtained from the first derivative of the 330/350 nm ratio plotted against temperature.

## Negative staining transmission electron microscopy

For negative staining transmission electron microscopy, p62$^{WT}$ or a mixture of p62$^{WT}$ and $VHH_{p62}$ 1:4 (12.5 µM in 50 mM HEPES pH 7.0, 250 mM NaCl, 5% Glycerol and 1 mM TCEP) were incubated for 2 h at 25 °C. The samples were diluted to 10–20 µg mL$^{-1}$ in buffer containing 50 mM HEPES pH 7.0, 250 mM NaCl, 5% Glycerol and 1 mM TCEP. 4 µL of diluted samples were adsorbed for 20 s onto glow-discharged Formvar/carbon-coated copper grids (Ted Pella, 01753-F). The grids were then blotted, washed with buffer and stained with 2% uranyl acetate solution. The grids were imaged using a Thermofisher Scientific Tecnai Spirit microscope operating at 120 kV. Electron micrographs were recorded using a FEI Veleta charge-coupled device camera at a pixel

size of 5.23 Å pixel$^{-1}$. P62 particles were analyzed using a custom pipeline in CellProfiler version 4.2.4.

## Nuclear magnetic resonance spectroscopy

Nuclear magnetic resonance spectroscopy was performed using a Bruker Avance III™ HD 600 MHz spectrometer equipped with $^1$H, $^{13}$C, $^{15}$N cryo-probes. ($^1$H,$^{15}$N)-SOFAST-HSQC spectra of 100 µM uniformly $^{13}$C,$^{15}$N-labeled ubiquitin (Silantes, 151603100) were recorded in the absence and presence of 100 µM $p62_{UBA}$, followed by addition of $VHH_{p62}$ to a final concentration of 120 µM. All measurements were performed in 25 mM HEPES pH 7.5, 150 mM NaCl, 1 mM TCEP, 10% (v/v) $D_2O$, 30 µM DSS at 296 K. All NMR spectra were processed and analyzed using Bruker TopSpin 3.2.1. All 1D $^1$H spectra were referenced by setting the DSS methyl peak to 0 ppm. ($^1$H,$^{15}$N)-SOFAST-HSQC spectra were referenced indirectly using the IUPAC recommended chemical shift referencing ratio (BMRB Entry 15410).

## X-ray structure determination and refinement of $VHH_{p62}$ and $p62_{UBA}$

$VHH_{p62}$ and p62$^{387-436}$ ($p62_{UBA}$) were mixed using a 1.2:1 molar ratio, incubated for 30 min on ice and concentrated up to 7.5 mg mL$^{-1}$. Sitting drop vapor diffusion crystallization was performed by pipetting 0.3 µL of protein complex and 0.3 µL of crystallization solution. Crystals appeared within 3–5 days with a size of 100–150 µm in a condition containing 1 M lithium chloride, 0.1 M citric acid, 20% (w/v) PEG6000. All crystallization trials were performed at 293 K. For data collection, the crystals were mounted onto cryoloops (Hampton Research) and flashcooled in liquid nitrogen. The crystals were measured in PXII at SLS.

Diffraction data were processed with autoPROC[66] (v1.1.7 Global Phasing Ltd) and Aimless[67] (v0.7.7). For structure determination, AlphaFold2[47] was used to generate a model of $VHH_{p62}$ (Supplementary Fig. S7F). The AlphaFold2 model was processed using Phenix process_predicted_model[68] (Phenix version v1.20.4459) to replace values in the B-factor field of the AlphaFold2 structure with calculated values. Values in the B-value field were interpreted as LDDT (0–100), and a maximum rmsd of 1.50 Å was used. The maximum B-value to be included was 59.22 Å$^2$, and a total of 116 of 121 residues were kept after B-factor filtering. Molecular replacement with Phaser[69] (v2.8.3), using coordinates of the processed $VHH_{p62}$ model, yielded a partial structure. The initial, partial solution was rebuilt with Coot[70] (v 0.9.8.7) and refined with Phenix. Molecular replacement was repeated with the rebuilt and refined partial model. The quality of the electron density map of the new solution allowed to trace and build a partial structure of the UBA domain of p62. The final structure was obtained by iteratively rebuilding and refining the coordinates of UBA and p62. Geometric correctness and Ramachandran plots were assessed using the Multimetric validation tool implemented in CCP4I2[71] (v.8.0.011). The final structure was submitted the RCSB Protein Data Bank: PDB ID: 9H1J. Details on data collection and refinement statistics are given in Table 1. All structure figures were generated using PyMOL (PyMOL Molecular Graphics System version 2.1.0; Schrödinger LLC).

## Prediction of the Ub:$p62_{UBA}$ complex

The complex between ubiquitin (NCBI: 9606, residues 1–76) and $p62_{UBA}$ (NCBI: 8878, residues 389–434) was calculated using a described pipeline in AlphaFold-Multimer v.3[47] and further processed using PyMOL (Schrödinger LLC) v.2.5.0.

## Protein expression and purification

***p62$^{WT}$ and p62$^{K7A/D69A}$.*** The constructs p62$^{WT}$ and p62$^{K7A/D69A}$ were expressed in *E. coli* BL21 (DE3) T1 resistant cells with an N-terminal His-tag followed by an IgG binding domain of protein A (ZZ-tag). Competent cells were transformed via heat shock with the respective

**Table 1 | Summary of data collection and refinement statistics**

| Protein | p62$_{UBA}$/VHH$_{p62}$ |
|---|---|
| PDB entry | 9H1J |
| Data collection | |
| Wavelength [Å] | 1.00007 |
| Space group | P2$_1$ |
| a, b, c [Å] | 62.9, 73.6, 89.8 |
| α, β, γ [°] | 90.0, 90.2, 90.0 |
| Resolution range [Å] | 19.89 – 1.73 |
| Unique reflections | 84831 |
| Mean I/sigma | 7.2 |
| Completeness | 0.99 |
| Redundancy | 5.3 |
| Rpim | 0.04 |
| CC (1/2) | 0.997 |
| Refinement | |
| Resolution range [Å] | 19.66 – 1.73 |
| R$_{work}$ | 0.2133 |
| R$_{free}$ | 0.2446 |
| *Number of non-hydrogen atoms* | |
| Macromolecule | 4951 |
| Solvent | 579 |
| *r.m.s deviation* | |
| Bond length (Å) | 0.0016 |
| Bond angles (°) | 0.380 |
| *Ramachandron plot*[a] | |
| Favored [%] | 96.66 |
| Allowed [%] | 3.34 |
| Generously [%] | 0 |

[a]Values from Multimetric validation tool in CCP4I2 (v.7.1.018).

plasmids and grown in Luria Bertani (LB) media containing 30 μg mL$^{-1}$ kanamycin overnight at 37 °C. Cells were then used for inoculation of 1 L Terrific Broth (TB) media containing 30 μg mL$^{-1}$ Kanamycin and grown at 37 °C until an OD$_{600nm}$ of 2.8–3.0. Protein expression was induced with 0.1 mM IPTG and carried out at 18 °C overnight. Cells were harvested by centrifugation (5000 × *g* at 4 °C for 20 min), flash frozen in liquid nitrogen, and subsequently stored at −80 °C.

Cell pellets were thawed on ice and resuspended in lysis buffer (50 mM HEPES pH 8.0, 500 mM NaCl, 2 mM MgCl$_2$, 20 mM Imidazole and 1 mM TCEP), together with cOmplete ULTRA EDTA-free protease inhibitor cocktail (Roche, 5892791001) and benzonase using a POLY-TRON PT1300 D handheld disperser (Kinematica). Cell lysis was carried out by sonication at 40% amplitude for 3 rounds of 1 min using a Sonics VCX 750 equipped with a standard probe (13 mm). The lysate was cleared by centrifugation at 40,000 × *g* for 1 h at 4 °C. An immobilized metal affinity chromatography (IMAC) step was performed by loading the cleared supernatant onto freshly equilibrated Talon resin (lysis buffer) and washing with lysis buffer before eluting the protein with elution buffer (50 mM HEPES pH 7.5, 500 mM NaCl, 2 mM MgCl$_2$, 500 mM Imidazole and 1 mM TCEP). The elution fraction was concentrated using an Amicon Ultra filter (30 kDa MWCO) and further purified by size exclusion chromatography (SEC) using a Superose® 6 Increase 10/300 GL column with SEC buffer (50 mM HEPES pH 7.0, 250 mM NaCl, 5% Glycerol and 1 mM TCEP).

**p62$_{UBA}$ and p62$_{ZZ}$.** The constructs His-MBP-p62$_{ZZ}$ and His-MBP-p62$_{UBA}$ were expressed in *Escherichia coli* BL21 (DE3) T1 resistant cells. Competent cells were transformed via heat shock with the respective plasmids and grown in Luria Bertani (LB) media containing

30 μg mL$^{-1}$ kanamycin overnight at 37 °C. Cells were then used for inoculation of 1 L Terrific Broth (TB) media containing 30 μg mL$^{-1}$ Kanamycin and grown at 37 °C until an OD$_{600nm}$ of 0.6–1.0. Protein expression was induced with 0.1 mM IPTG and performed at 18 °C overnight. Cells were harvested by centrifugation (5000 × *g* at 4 °C for 20 min), and subsequently stored at −80 °C.

Cell pellets were thawed on ice and resuspended in lysis buffer (50 mM Tris pH 7.0, 300 mM NaCl, 2 mM MgCl$_2$, 20 mM Imidazole, 10% glycerol and 1 mM TCEP), together with cOmplete ULTRA EDTA-free protease inhibitor cocktail (Roche, 5892791001) and benzonase. Cell lysis was carried out using a Microfluidizer (Microfluidics) for 3 rounds at 20,000 psi. The lysate was cleared by centrifugation at 30,000 × *g* for 30 min at 4 °C. An immobilized metal affinity chromatography (IMAC) step was performed by loading the cleared supernatant onto freshly equilibrated HisTrap FF resin (lysis buffer) and washing with lysis buffer before eluting the protein using an Imidazole gradient in elution buffer (50 mM Tris pH 7.0, 300 mM NaCl, 300 mM Imidazole, 10% glycerol and 1 mM TCEP). The His-MBP tag was cleaved using 3C protease. The cleavage reaction was incubated overnight at 4 °C and then concentrated using an Amicon Ultra filter (3 kDa MWCO) and further purified by size exclusion chromatography (SEC) using a Superdex® 75 16/60 HiLoad column with SEC buffer (20 mM Tris pH 8.0, 150 mM NaCl and 1 mM TCEP).

**VHH$_{p62}$-His (6C10).** The construct VHH$_{p62}$-His (6C10) was cloned into the *Pichia pastoris* vector backbone followed by a linearization step using SacI enzyme. Competent *Pichia pastoris (Komatagaella phaffii)* cells (NRRL-Y11430 wildtype) were subsequently transformed via electroporation of the linearized DNA and grown in YPD/sorbitol broth media at 30 °C for 6 h. Aliquots of recovered cells were spread onto YPD agar plates containing 100 mg/L$^{-1}$ zeocine. After 3 days incubation at 30 °C in the absence of light, single colonies were picked and used to inoculate 3 mL BMGY broth for expression at 30 °C and 230 rpm shaking in 24 well plates. 24 h post inoculation, expression cultures were harvested by centrifugation (4000 × *g*, 15 min, 4 °C) and induced via media switch to 3 mL BMMY broth. Induction was maintained by addition of 2% MeOH to expression cultures 24 h and 48 h post induction. Culture supernatants were harvested by centrifugation 48 h and 72 h post induction, and expression titer was determined by ALC-HPLC and on LabChip GXII HT (PerkinElmer) to select the best production clone. The selected expression clone was pre-cultivated in 100 mL YPD broth for 20 h at 30 °C and 230 rpm shaking. 2% of selected clone pre-culture was used to inoculate 300 mL of BMGY broth at 30 °C and 230 rpm shaking in 1 L baffled Erlenmeyer flasks. 24 h post inoculation, cells were harvested via centrifugation (4000 × *g*, 20 min, 4 °C) and induced via media switch to BMMY broth (300 mL culture volume per flask). Expression cultures were supplemented with 2% MeOH 24 h and 48 h post induction. 72 h post induction, the expression culture was harvested via centrifugation (4500 × *g*, 20 min, 4 °C). The harvested medium supernatant was sterile filtered before further processing. An immobilized metal affinity chromatography (IMAC) step was performed by loading the titrated and filtrated supernatant onto pre-equilibrated His trap excel resin column with 50 mM NaH$_2$PO$_4$, 500 mM NaCl, 20 mM Imidazole, pH 7.4 buffer (equilibration buffer) and washed with equilibration buffer before eluting the VHH$_{p62}$-His protein with elution buffer (50 mM NaH$_2$PO$_4$, 500 mM NaCl, 500 mM Imidazole, pH 7.4). Buffer exchange was subsequently performed on the eluted fraction using HiPrep Desalting column, pre-equilibrated with PBS and further concentrated using stirring cells with a membrane cut-off of 1 KDa.

## Reporting summary
Further information on research design is available in the Nature Portfolio Reporting Summary linked to this article.

## Data availability

All data supporting the findings of this study are available within the paper, the Supplementary Information, and the Source Data file. The X-ray crystallography data generated in this paper have been deposited in the RCSB Protein Data Bank with the PDB ID: 9H1J. The affinity proteomics data generated in this paper have been deposited in the PRIDE databank under accession code PXD057810. Source data are provided with this paper.

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

## Acknowledgements

We thank Isabelle Claerr, for imaging support. We thank Seth Carbonneau for the discussion about VHH$_{GFP}$-based proximity strategies and for providing the plasmid encoding SPOP_BTB. We thank Anke Harupa for providing mentorship during the execution of this study, and Anne Granger and Jonathan Solomon for the critical review of this manuscript. Open Innovation at Global Health DA was supported by a grant from the Gates Foundation (Thierry T. Diagana; Grant no. INV010720). The conclusions and opinions expressed in this work are those of the authors alone and shall not be attributed to the Foundation.

## Author contributions

Z.T, D.M., G.M. and B.N. conceived the idea, designed experiments and interpreted the data. C.M., N.S. and G.S. performed cloning and cell biology experiments. R.M. led the efforts in library generation and gateway cloning experiments. B.G. and L.E. led the efforts in protein biochemistry experiments. B.G., L.E., C.F. and S.C. performed the NMR studies. A.H., N.G., M.V., S.S., A.L., J-Y.C., L.F., P.H., C.B., M.M., F.F. and M.P. contributed to the efforts in generation, production, and characterization of VHH$_{p62}$. C.S.M., A.H. and S.G. performed mass spectrometry and data analysis for affinity proteomics experiments. A.L. contributed to the efforts in generation of ATG5 K.O. cell lines. F.V. and S.G. led the efforts in resolving VHH$_{p62}$:p62$_{UBA}$. G.M., L.T. and A.P. designed and conducted induced proximity experiments in Huh7 cells. Z.T. and B.N. wrote the manuscript with critical review from G.M. and G.S.

## Competing interests

All authors were employed by and/or shareholders of Novartis Pharma AG during this study.
