## [Transparent Peer Review file · Nature Communications]

Development of a p62 biodegrader for autophagy targeted degradation

Corresponding Author: Dr Beat Nyfeler

Version 0:

Reviewer comments:

Reviewer #1

(Remarks to the Author)

Autophagy degrades various intracellular components selectively by using autophagy cargo receptors such as p62, NDP52, NBR1, and OPTN. These receptors oligomerize and recruit ATG8-family proteins to the cargo, facilitating the formation of autophagosomes around the cargo. The versatility of lysosomes to degrade large and complex cargo makes autophagy a promising target for therapeutic interventions. Proteasome-based targeted protein degradation (TPD) has emerged as a therapeutic modality, relying on molecules like PROTACs to induce proximity between a target and an E3 ligase, leading to the target's degradation by the proteasome. However, this approach is limited to substrates accessible to the proteasome. Alternative strategies utilizing the autophagy-lysosome pathway have been developed, expanding the substrate scope to include protein aggregates and organelles. These strategies involve various autophagy targeting effectors, such as autophagy cargo receptors (AUTOTACs), ATG8-family proteins (ATTECs), and kinases (ULKRECs). However, it remains unclear which components of the autophagy machinery are most effective for inducing selective target degradation. This study systematically evaluated 93 autophagy effector candidates by artificially recruiting them to mitochondria and monitoring mitophagy activity and identified autophagy cargo receptors (p62, NBR1, NDP52), ATG8 family proteins, and two kinases (ULK1 and TBK1) that induce mitophagy. The authors then studied the targeted degradation of a soluble protein by these effectors and found that self-oligomerizing autophagy cargo receptors outperform other autophagy effectors. Finally, the authors developed a p62 biodegrader for autophagy targeted degradation by obtaining anti-p62 VHH and fusing it to VHHGFP and showed that VHHGFP-VHHp62 promoted autophagic degradation of mito-mCh-GFP but not cyto-mCh-GFP, concluding that this degrader can induce autophagy-dependent degradation of scaffolded targets such as organelles but not soluble proteins.

TPD is a drug discovery technology that is currently attracting a great deal of attention, and TPD utilizing autophagy, which includes organelles and aggregates as degradation targets, is particularly important as a future technology. Although a number of autophagy-based TPDs have been reported in the past, this paper for the first time provides a comprehensive analysis of which effectors are more efficient and provides a foundation for the development of autophagy-based TPDs. Moreover, by comparing autophagic degradation of mitochondria and a soluble protein, this paper also showed that organelles and soluble proteins require different mechanisms for autophagy-based degradation. The experiments are well designed, and the obtained results are convincing. Thus, this paper greatly contributes to the field of TPDs utilizing autophagy. On the other hand, only mitochondria and monomeric proteins have been examined as targets, and soluble multimeric proteins that fall in between should also be examined. Furthermore, analysis and discussion should take into account the phenomenon of p62 phase separation with poly-ubiquitin chains.

Major points

- 1) It is known that the p62 PB1 domain promotes self-oligomerization whereas the p62 UBA domain interacts with ubiquitin. The authors showed that VHHp62 binds to the UBA domain of p62 by X-ray crystallography and SPR. Then, why did VHHp62 inhibit self-oligomerization of p62? VHHp62 might also interact with the PB1 domain and inhibit its oligomerization. Perform SPR between VHHp62 and p62 PB1 domain.
- 2) It is known that p62 undergoes phase separation via interaction with poly-ubiquitin chains to form p62 bodies (droplets), which are targeted by selective autophagy. Because VHHp62 competes with ubiquitin for binding to the UBA domain of p62, VHHGFP-VHHp62 would inhibit the phase separation of p62. The observation that VHHGFP-VHHp62 did not promote autophagic degradation of cytosolic soluble protein may be due to the inhibition of p62 phase separation. Design a degrader that binds p62 and targets without inhibiting the p62-Ub interaction and find out if it can degrade soluble proteins.
- 3) The authors concluded that VHHGFP-VHHp62 induces autophagic degradation of scaffolded targets but not soluble

proteins by studying the degradation of mito-mCh-GFP as a scaffolded target and cyt-mCh-GFP as a soluble target. Because cyt-mCh-GFP is monomeric, it is not clear whether VHHGFP-VHHp62 can induce autophagic degradation of soluble, but multimeric targets. Design soluble, multimeric mCh-GFP and study whether VHHGFP-VHHp62 can degrade it or not.

Minor points

1) Line 168, "ATG-like proteins" should be "ATG8-like proteins".

Reviewer #2

(Remarks to the Author)

This manuscript identified effector proteins that efficiently drive target degradation by autophagy. The authors developed a VHH-based target degradation technique in which mCherry-GFP-fused substrates are targeted by GFP nanobody-fused autophagy effectors. Among the effectors they screened, autophagy adaptors such as p62 and NBR1, ATG8 family proteins, and kinases such as ULK1 and TBK1 efficiently cleared mitochondria by autophagy. Autophagy adaptors also efficiently caused degradation of a soluble protein (mCherry-GFP). The authors showed that the self-oligomerizing PB1 domain and ATG8-binding LIR motif were the minimal components sufficient to drive target degradation of mitochondria and cytosolic proteins. They further expanded this VHH-based approach by developing a nanobody against p62. Although this p62 nanobody had an inhibitory effect on p62 oligomerization and p62 VHH-GFP VHH fusion degrader was unable to degrade cytosolic protein, it efficiently recruited endogenous p62 to mitochondria and led their degradation. The experiments are well-designed, and the manuscript is well-written. Their findings provide important insights to the research field of target degradation. This reviewer only has a couple of comments to strengthen their findings.

1. To confirm that VHH-based autophagy effectors recruit the autophagy machinery, they should show that autophagosomes are recruited to the target sites by staining with LC3 in Figures 1C, 3B, 3I, and 5A.
2. To further validate that VHH-based autophagy effectors degrade mitochondria and soluble proteins via autophagy, the authors should show that knockdown or knockout of a core autophagy gene cancels the degradation of these target substrates in Figures 1C, 1D, 3B, 3D, 3I, 3K, 5A, and 5C.
3. To further characterize the functionality of p62 VHH-based degrader, the authors should expand repertoire of the substrates that are degraded by this degrader (e.g. protein aggregates).
4. As the p62 VHH-based degrader degrades targets by recruiting ATG8s, it is important to show p62 VHH immunoprecipitates contain ATG8s in Figure 4G.
5. The MS data is generally reliable, and the results are clear. However, it remains unclear whether the observed binding is weak or simply background binding. Additionally, since the comparison is based on enrichment relative to whole cells, it appears to exclude proteins that could not be identified in whole-cell analysis (Supplementary Table 2). Given the low copy number of autophagy-related factors, there is a possibility that important factors such as LC3 and Atg8 might be excluded. What would the results show if the factors binding to VHHp62 were compared between Huh7 and Huh7 p62 KO cells?

Reviewer #3

(Remarks to the Author)

In their study Development of a p62 biodegrader for autophagy targeted degradation Thiel and colleagues screened for factors that could be used for targeting mitochondria or cytosolic proteins to lysosomal degradation. Identifying p62 as one of the most efficient factor, the authors analyzed which domains and characteristics of p62 are essential and sufficient for this effect: clustering (oligomerization) and avidity. The authors subsequently raised a nanobody against p62's UBA domain and generated a bifunctional nanobody pair against GFP and p62, which can serve as a biodegrader for GFP-labeled mitochondria but not for a cytosolic GFP-mCherry construct.

While I find the study overall interesting, well structured, and technically of high quality, it in this stage lacks sufficient novelty for publication in Nat Comm. Major points for this assessment are:

1) The authors describe their screening setup, which is very similar to the one that has been described by <https://doi.org/10.1080/15548627.2024.2395149> and this work should have also been mentioned and cited. In particular, because this published work has already reported 5 of the 9 mitophagy inducing factors (ULK1, GAB, GABL2, LC3A, LC3B) described in the current study.

On the other hand, one has to acknowledge that this current study has screened a much larger library of effector candidates (93). This (largely negative) data is one of the valuable pieces of data within this study and surely worth to be published in general. Also, p62, the effector that was chosen for the development of a potential biodegrader, hasn't been reported in this context before.

2) The authors report that efficient autophagic degradation via the p62 effector relies on avidity and oligomerization. This concept is not really new and is well established for p62's physiological functions. The avidity experiment with several LIR

motifs behind each other / oligomerization via PB1 domain to improve LC3 / autophagic membrane recruitment is similar to analysis/findings made in <https://doi.org/10.15252/emboj.201695063>.

The direct comparison of the effect on mito-mCh-GFP and cyt-mCh-GFP is however interesting.

3) The authors decided to generate a VHH(p62)-VHH(GFP) biodegrader for validation. As such, the effect of the biodegrader shown in the manuscript depends on an artificial double-overexpression system (biodegrader + mito-mCh-GFP).

It is highly likely that only the strong overexpression and therefore very high abundance of GFP on mitochondria is enabling an efficient degradation of mitochondria. While suggested biodegraders may have high potential, it remains questionable, if the abundance of any endogenous protein (even though highly expressed on e.g. mitos like VDAC1, TOM20, ...) would be sufficient to allow targeted degradation of cellular content via the autophagic system. In the absence of this proof of concept, the applicability of the presented biodegraders remains very vague.

4) The presented effects on cytosolic targets (no degradation) were predictable, given the large amount of literature on p62 function in aggrephagy (where also NBR1 belongs).

As such, the authors in large parts rather confirmed (in very high quality and combined within a single manuscript) several known aspects of p62 function and aggrephagy, while failing to provide real evidence that their developed biodegrader is functional in a physiological setting. Providing data on the successful degradation of any cellular target without the need of extensive (surface) labelling (by GFP) would provide sufficient novelty for publication in Nat Comm. Otherwise, another journal may be more suitable for publication.

Other major points:

- Fig 1D: would be good to show at one point -/+Baf control for autophagic degradation.
- Fig. 3B: Gabarap seems to work better than NBR1. Text and 3C seem to contradict each other. Maybe reanalyze same images with total integrated Green/Red signal as ratiometric readout.
- Fig. S4B –shouldn't there be any mCh+/GFP- dots, because Baf should prevent the acidification of lysosomes? What about WB with -/+ Baf in addition?
- Text 168-170: Where can one see the large structures? In general, more frequent indications where to find the respective data would have been helpful.
- Text 171-173 / Fig 3F: Δ PB1 mutant was significantly less potent in clearing (50%) and not "almost as effective". Needs to be corrected. The fact that the activity of the point mutation is higher than Δ PB1 mutant may indicate remaining oligomerization properties and that probably endogenous p62 is contributing a lot to the clearance. KD of endogenous p62 needed to be able to evaluate this, because the LIR is probably as important and it is mostly/only endogenous p62-LIR that preserves functionality of this mutation. Since the authors now do have a p62 KO cell line, this specific experiment could be reproduced under clean p62 KO conditions.
- Fig. 4G: There seems to be a quite high background of binding to beads. What did the authors consider as robust enrichment and why not plotting IP WT/KO against each other? That should give a more specific picture.
- Text 250ff – is the effect seen rather sth similar to a Hook effect than preventing oligomerization? Too much dilution by forming individual dimers rather than oligomers? There is literature out there that PB1 and UBA domain are contributing to the formation of larger oligomers. Using a construct that is not limited to ZZ domain but includes the PB1 domain in Fig. 4C could further clarify this point as well.

Other minor points:

- Fig 3D: why does the mCh signal disappears while GFP signal partly stays? Really from the same blot or potentially two different ones with issues in sample transfer?
- Transient and stable expression should be indicated/defined for each construct in all figure legends.
- The terms "selective" and "potent" should be used carefully according to their actual definition.

Reviewer #4

(Remarks to the Author)

Version 1:

Reviewer comments:

Reviewer #1

(Remarks to the Author)

The authors have carried out extensive experiments, and it is highly commendable that they demonstrated the degrader can also break down multimeric reporter proteins. Overall, the manuscript has been substantially strengthened.

Reviewer #2

(Remarks to the Author)

The authors have adequately addressed all of our concerns through both additional experiments and explanatory comments.

They have provided new data that strengthen their conclusions as well as reasonable explanations.

However, the authors only included quantification of microscopic images in Figures S2B, S2C, S4C, and S10B.

They should also include representative images corresponding to these quantifications.

Reviewer #3

(Remarks to the Author)

The authors have addressed my concerns sufficiently to recommend publication in Nature Communications. I congratulate them on this nice piece of work.

Reviewer #4

(Remarks to the Author)

Reviewer #1 (Remarks to the Author):

Autophagy degrades various intracellular components selectively by using autophagy cargo receptors such as p62, NDP52, NBR1, and OPTN. These receptors oligomerize and recruit ATG8-family proteins to the cargo, facilitating the formation of autophagosomes around the cargo. The versatility of lysosomes to degrade large and complex cargo makes autophagy a promising target for therapeutic interventions. Proteasome-based targeted protein degradation (TPD) has emerged as a therapeutic modality, relying on molecules like PROTACs to induce proximity between a target and an E3 ligase, leading to the target's degradation by the proteasome. However, this approach is limited to substrates accessible to the proteasome. Alternative strategies utilizing the autophagy-lysosome pathway have been developed, expanding the substrate scope to include protein aggregates and organelles. These strategies involve various autophagy targeting effectors, such as autophagy cargo receptors (AUTOTACs), ATG8-family proteins (ATTECs), and kinases (ULKRECs). However, it remains unclear which components of the autophagy machinery are most effective for inducing selective target degradation. This study systematically evaluated 93 autophagy effector candidates by artificially recruiting them to mitochondria and monitoring mitophagy activity and identified autophagy cargo receptors (p62, NBR1, NDP52), ATG8 family proteins, and two kinases (ULK1 and TBK1) that induce mitophagy. The authors then studied the targeted degradation of a soluble protein by these effectors and found that self-oligomerizing autophagy cargo receptors outperform other autophagy effectors. Finally, the authors developed a p62 biodegrader for autophagy targeted degradation by obtaining anti-p62 VHH and fusing it to VHHGFP and showed that VHHGFP-VHHp62 promoted autophagic degradation of mito-mCh-GFP but not cyto-mCh-GFP, concluding that this degrader can induce autophagy-dependent degradation of scaffolded targets such as organelles but not soluble proteins. TPD is a drug discovery technology that is currently attracting a great deal of attention, and TPD utilizing autophagy, which includes organelles and aggregates as degradation targets, is particularly important as a future technology. Although a number of autophagy-based TPDs have been reported in the past, this paper for the first time provides a comprehensive analysis of which effectors are more efficient and provides a foundation for the development of autophagy-based TPDs. Moreover, by comparing autophagic degradation of mitochondria and a soluble protein, this paper also showed that organelles and soluble proteins require different mechanisms for autophagy-based degradation. The experiments are well designed, and the obtained results are convincing. Thus, this paper greatly contributes to the field of TPDs utilizing autophagy. On the other hand, only mitochondria and monomeric proteins have been examined as targets, and soluble multimeric proteins that fall in between should also be examined. Furthermore, analysis and discussion should take into account the phenomenon of p62 phase separation with poly-ubiquitin chains.

We would like to thank reviewer #1 who agrees that our study is well designed and that our results are convincing but points out that the scope of targets explored is too narrow and only includes mitochondria and monomeric proteins. As explained in detail below, we have expanded the repertoire of targets by including additional multimeric and oligomeric cytosolic reporters (Supplementary Figure S8). Furthermore, we now discuss the phenomenon of p62 phase separation with poly-ubiquitin chains in our discussion (see line 349-352).

Major points

1) It is known that the p62 PB1 domain promotes self-oligomerization whereas the p62 UBA domain interacts with ubiquitin. The authors showed that VHHp62 binds to the UBA domain of p62 by X-ray crystallography and SPR. Then, why did VHHp62 inhibit self-oligomerization of p62? VHHp62 might also interact with the PB1 domain and inhibit its oligomerization. Perform SPR between VHHp62 and p62 PB1 domain.

Testing recombinant PB1 via SPR is challenging due to its tendency to aggregate. Since the p62 immunogen in our Llama immunization campaign lacked the PB1 domain, it is highly unlikely that VHH_{p62} recognizes the PB1 domain via a high affinity interaction. We adjusted the text and figure 4 A to clarify that the immunogen did not contain the PB1 domain.

2) It is known that p62 undergoes phase separation via interaction with poly-ubiquitin chains to form p62 bodies (droplets), which are targeted by selective autophagy. Because VHHp62 competes with ubiquitin for binding to the UBA domain of p62, VHHGFP-VHHp62 would inhibit the phase separation of p62. The observation that VHHGFP-VHHp62 did not promote autophagic degradation of cytosolic soluble protein may be due to the inhibition of p62 phase separation. Design a degrader that binds p62 and targets without inhibiting the p62-Ub interaction and find out if it can degrade soluble proteins.

We agree that VHH_{p62} may inhibit phase separation of p62, as evidenced by our data in Figure S9 A-C, where no p62 punctae formation was observed in the presence of VHH_{p62}. This inhibition likely contributes to the lack of autophagic degradation of cytosolic soluble proteins observed with VHH_{GFP}-VHH_{p62}. Our immunization campaign did not identify VHHS that bind to p62 outside the UBA domain. While developing a new immunization campaign is beyond the scope of our current study, we acknowledge the potential of new screens for additional p62 binders.

3) The authors concluded that VHHGFP-VHHp62 induces autophagic degradation of scaffolded targets but not soluble proteins by studying the degradation of mito-mCh-GFP as a scaffolded target and cyt-mCh-GFP as a soluble target. Because cyt-mCh-GFP is monomeric, it is not clear whether VHHGFP-VHHp62 can induce autophagic degradation of soluble, but multimeric targets. Design soluble, multimeric mCh-GFP and study whether VHHGFP-VHHp62 can degrade it or not.

We would like to thank the reviewer for this important recommendation. The revised manuscript describes two additional reporters: 1) cytosolic HaloTag-mCh-GFP-SNAPtag which can be multimerized upon treatment with HaXS8, and 2) coilin_{NTD}-mCh-GFP which contains the N-terminal domain of coilin and forms oligomeric structures. While multimerization of HaloTag-mCh-GFP-SNAPtag was not sufficient to trigger degradation, VHH_{GFP}-VHH_{p62} efficiently triggered lysosomal delivery and degradation of coilin_{NTD}-mCh-GFP. These new results strengthen the conclusions of our study and are shown in Figure S8 and are discussed in lines 277-286 of the manuscript.

Minor points

Line 168, "ATG-like proteins" should be "ATG8-like proteins".

Thank you. We corrected the text accordingly.

Reviewer #2 (Remarks to the Author):

This manuscript identified effector proteins that efficiently drive target degradation by autophagy. The authors developed a VHH-based target degradation technique in which mCherry-GFP-fused substrates are targeted by GFP nanobody-fused autophagy effectors. Among the effectors they screened, autophagy adaptors such as p62 and NBR1, ATG8 family proteins, and kinases such as ULK1 and TBK1 efficiently cleared mitochondria by autophagy. Autophagy adaptors also efficiently caused degradation of a soluble protein (mCherry-GFP). The authors showed that the self-oligomerizing PB1 domain and ATG8-binding LIR motif were the minimal components sufficient to drive target degradation of mitochondria and cytosolic proteins. They further expanded this VHH-based approach by developing a nanobody against p62. Although this p62 nanobody had an inhibitory effect on p62 oligomerization and p62 VHH-GFP VHH fusion degrader was unable to degrade cytosolic protein, it efficiently recruited endogenous p62 to mitochondria and led their degradation. The experiments are well-designed, and the manuscript is well-written. Their findings provide important insights to the research field of target degradation. This reviewer only has a couple of comments to strengthen their findings.

We would like to thank reviewer #2 who agrees that our study is well designed and that our manuscript is well written. As explained in detail below, we have performed further experiments to show dependence of degradation on autophagy and expanded the repertoire of targets by including additional multimeric and oligomeric cytosolic reporters and a system to degrade damaged mitochondria (Figure 5, Figure S8, and Figure S10).

1. To confirm that VHH-based autophagy effectors recruit the autophagy machinery, they should show that autophagosomes are recruited to the target sites by staining with LC3 in Figures 1C, 3B, 3I, and 5A.

We were unable to reliably detect endogenous LC3 and GABARAP by immunofluorescence despite our efforts in testing different antibodies. To strengthen ATG8-dependence of VHH-based autophagy effectors, we expressed HaloTag-MAP1LC3B in HeLa mito-mCh-GFP cells with inducible expression of VHH_{GFP}, VHH_{GFP}-p62 and VHH_{GFP}-VHH_{p62}. Staining with a HaloTag-specific fluorophore confirmed co-localization of LC3B with the mito-mCh-GFP reporter when VHH_{GFP}-p62 and VHH_{GFP}-VHH_{p62} were expressed. While the data is included below, we believe presenting these results in the main manuscript does not significantly enhance its value. Instead, we included in the revised manuscript new data with the VPS34 inhibitor PIK-III and generated ATG5 K.O. cells to substantiate autophagy-dependence of VHH-based autophagy effectors (see major point #2 below).

2. To further validate that VHH-based autophagy effectors degrade mitochondria and soluble proteins via autophagy, the authors should show that knockdown or knockout of a core autophagy gene cancels the degradation of these target substrates in Figures 1C, 1D, 3B, 3D, 3I, 3K, 5A, and 5C.

We would like to thank reviewer #2 for this suggestion that has strengthened the conclusions of our study. In addition to experiments with Bafilomycin A1, we now show that the VPS34 inhibitor PIK-III robustly inhibits lysosomal translocation of mitochondrial as well as cytosolic mCh-GFP upon expression of VHH-based autophagy effectors. To further address this comment, we generated HeLa mito-mCh-GFP ATG5 K.O. cells and show that VHH_{GFP}-p62 and VHH_{GFP}-VHH_{p62} can no longer degrade mitochondria in these cells, validating dependence on the autophagy core machinery. This new data is presented in Figure S2 B-D, Figure S4 C, and Figure S10 B-D.

3. To further characterize the functionality of p62 VHH-based degrader, the authors should expand repertoire of the substrates that are degraded by this degrader (e.g. protein aggregates).

The revised manuscript describes two additional substrates and shows that VHH_{GFP}-VHH_{p62} efficiently drives lysosomal delivery and degradation of oligomeric coilin_{NTD}-mCh-GFP as described above for major point 3 of reviewer #1. Furthermore, we expand the functionality of the p62 VHH by developing a degrader that selectively recognizes damaged mitochondria. The PINK1-VHH_{p62} degrader is now described in Figure 5 and Supplementary Figure 10 and strengthens the importance of avidity for autophagy targeted degradation. Avidity of PINK1-VHH_{p62} is only provided upon accumulation of the degrader on the surface of depolarized mitochondria which in turn drives the recruitment of endogenous p62 and subsequent lysosomal delivery.

4. As the p62 VHH-based degrader degrades targets by recruiting ATG8s, it is important to show p62 VHH immunoprecipitates contain ATG8s in Figure 4G.

We thank reviewer #2 for raising this point. We recognize that the text in the original manuscript may not have clearly conveyed our intended message, potentially leading to confusion. The

immunoprecipitation study presented in Figure 4G was performed with the goal of characterizing the specificity of VHH_{p62} for the UBA domain of p62. We therefore chose stringent cell lysis conditions using RIPA buffer. While some interactions partners of p62 were co-immunoprecipitated, we did not detect ATG8 family members. In addition to the stringent cell lysis conditions, we likely missed interactions with ATG8-family members since VHH_{p62} may inhibit phase separation of p62 when used in the absence of a scaffolded substrate (see major point #2 of reviewer #1).

5. The MS data is generally reliable, and the results are clear. However, it remains unclear whether the observed binding is weak or simply background binding. Additionally, since the comparison is based on enrichment relative to whole cells, it appears to exclude proteins that could not be identified in whole-cell analysis (Supplementary Table 2). Given the low copy number of autophagy-related factors, there is a possibility that important factors such as LC3 and Atg8 might be excluded. What would the results show if the factors binding to VHHp62 were compared between Huh7 and Huh7 p62 KO cells?

We would like to thank the reviewer for carefully assessing the MS data and would like to clarify that the primary goal of the MS experiment was to assess the selectivity of VHH_{p62} rather than identify binding partners of p62 (see major point #4 above). VHH_{p62} was not expressed in cells but only added as an affinity reagent for the pull-down step. Furthermore, lysates from Huh7 and Huh7 p62 K.O. cells were processed in separate MS runs making a differential analysis between these two experiments not reliable. We adjusted Figure 4G to specifically highlight proteins with UBA domains to clarify the primary goal of the study and hope this clarification addresses any concerns.

Reviewer #3 (Remarks to the Author):

In their study Development of a p62 biodegrader for autophagy targeted degradation Thiel and colleagues screened for factors that could be used for targeting mitochondria or cytosolic proteins to lysosomal degradation. Identifying p62 as one of the most efficient factor, the authors analyzed which domains and characteristics of p62 are essential and sufficient for this effect: clustering (oligomerization) and avidity. The authors subsequently raised a nanobody against p62's UBA domain and generated a bifunctional nanobody pair against GFP and p62, which can serve as a biodegrader for GFP-labeled mitochondria but not for a cytosolic GFP-mCherry construct.

While I find the study overall interesting, well structured, and technically of high quality, it in this stage lacks sufficient novelty for publication in Nat Comm. Major points for this assessment are:

1) The authors describe their screening setup, which is very similar to the one that has been described by <https://doi.org/10.1080/15548627.2024.2395149> and this work should have also been mentioned and cited. In particular, because this published work has already reported 5 of the 9 mitophagy inducing factors (ULK1, GAB, GABL2, LC3A, LC3B) described in the current study.

On the other hand, one has to acknowledge that this current study has screened a much larger library of effector candidates (93). This (largely negative) data is one of the valuable pieces of data within this study and surely worth to be published in general. Also, p62, the effector that was chosen for the development of a potential biodegrader, hasn't been reported in this context before.

We would like to thank reviewer #3 for the valuable feedback and for pointing out the additional reference which is now included as citation in line 90 and discussed in lines 321-323 of the manuscript. We agree with the reviewer that our work distinguishes itself by utilizing a significantly larger library, which enabled us to identify factors that were not mentioned by Lorentzen et al. We believe that the partial overlap between the two studies validates the robustness of our experimental approach in identifying relevant effectors for targeted autophagy degradation. The discovery of VHH_{p62} and its application as an autophagy degrader is highly novel. In the revised manuscript, we now describe the engineering of VHH_{p62} into a PINK1-based biodegrader which can selectively recognize and target damaged mitochondria. This new data hopefully addresses concerns regarding sufficient novelty for publication in Nature Communications.

2) The authors report that efficient autophagic degradation via the p62 effector relies on avidity and oligomerization. This concept is not really new and is well established for p62's physiological functions. The avidity experiment with several LIR motifs behind each other / oligomerization via PB1 domain to improve LC3 / autophagic membrane recruitment is similar to analysis/findings made in <https://doi.org/10.15252/embj.201695063>.

The direct comparison of the effect on mito-mCh-GFP and cyt-mCh-GFP is however interesting.

Thank you for pointing out the additional reference which is now included as citation in line 192 of the manuscript. While the referenced study provides important insights into avidity for autophagy, it did not focus on the concept of targeted degradation or induced proximity approaches. Our study extends these findings by specifically demonstrating how avidity and oligomerization principles can be harnessed to achieve targeted autophagic degradation. We agree with the reviewer that our study differentiates itself by directly and more systematically comparing the concept across different substrates. The revised manuscript contains two additional reporters (HaloTag-mCh-GFP-SNAPtag and coilin_{NTD}-mCh-GFP) which allowed us to highlight the versatility and limitations of avidity-induced degradation (Figure S8).

3) The authors decided to generate a VHH(p62)-VHH(GFP) biodegrader for validation. As such, the effect of the biodegrader shown in the manuscript depends on an artificial double-overexpression system (biodegrader + mito-mCh-GFP). It is highly likely that only the strong overexpression and therefore very high abundance of GFP on mitochondria is enabling an efficient degradation of mitochondria. While suggested biodegraders may have high potential, it remains questionable, if the abundance of any endogenous protein (even though highly expressed on e.g. mitos like VDAC1, TOM20, ...) would be sufficient to allow targeted degradation of cellular content via the autophagic system. In the absence of this proof of concept, the applicability of the presented biodegraders remains very vague.

We acknowledge the concern that artificial overexpression of GFP at the mitochondrial membrane may drive the effectiveness of VHH_{GFP}-VHH_{p62} to induce mitophagy. Unfortunately, we are not aware of any currently available VHHs targeting endogenous mitochondrial membrane proteins, which could be used to generate a bi-functional biodegrader and to address this concern, and generating such VHHs de novo is beyond the scope of this study. However, as a surrogate approach, we describe in the revised manuscript a PINK1-VHH_{p62} degrader which does not rely on the binding of highly abundant GFP on the mitochondrial membrane. In this setup, the mCh-GFP tag is only used to visualize the delivery of mitochondria to lysosomes. Expression of PINK1-VHH_{p62} allowed the selective recruitment of endogenous p62 to uncoupled mitochondria and their delivery

to lysosomes. This result suggests that VHH_{p62} degraders can induce mitophagy in the absence of an artificial double-overexpression system and expand the scope of this study. These findings further highlight the potential of autophagy-targeted degradation technologies for selectively removing damaged mitochondria, adding significant value and novelty to the manuscript.

4) The presented effects on cytosolic targets (no degradation) were predictable, given the large amount of literature on p62 function in aggrephagy (where also NBR1 belongs).

We appreciate the provided perspective as the lack of degradation of cytosolic targets was not predictable to us. We initially demonstrated that VHH_{GFP-p62}, as well as a minimal construct combining the PB1 domain with a LIR motif, is sufficient to induce degradation of cytosolic targets. These findings align with literature stating that p62 can drive autophagic clearance of soluble, non-aggregated substrates. Hence, we did not expect the observation that VHH_{p62}-mediated recruitment of endogenous p62 does not result in degradation of a cytosolic target. Our follow-up investigations revealed that this effect arises because VHH_{p62} inhibits the oligomerization of p62, as shown by our experiments in cells and with recombinant proteins (Figure S9).

As such, the authors in large parts rather confirmed (in very high quality and combined within a single manuscript) several known aspects of p62 function and aggrephagy, while failing to provide real evidence that their developed biodegrader is functional in a physiological setting. Providing data on the successful degradation of any cellular target without the need of extensive (surface) labelling (by GFP) would provide sufficient novelty for publication in Nat Comm. Otherwise, another journal may be more suitable for publication.

We are not aware of any reported binders of p62 that can be expressed in cells and applied as autophagy biodegraders. We therefore believe that the discovery of VHH_{p62} provides an important research tool to test the therapeutic potential of autophagy degraders. As described for major point #3 above, the revised manuscript now describes a PINK1-based biodegrader which does not rely on the extensive surface labelling by GFP.

Other major points:

- Fig 1D: would be good to show at one point +/-Baf control for autophagic degradation.

We monitored autophagic degradation in this study over a time period of 72 h which is not compatible with Bafilomycin A1 treatment due to cell toxicity. The effect on mCh⁺/GFP⁺ punctae formation was analyzed by adding Bafilomycin A1 for the last 15 h. However, the revised manuscript contains data in ATG5 K.O. cells where autophagic degradation was blocked (Figures S2 D and S10 D).

- Fig. 3B: Gabarap seems to work better than NBR1. Text and 3C seem to contradict each other. Maybe reanalyze same images with total integrated Green/Red signal as ratiometric readout.

We re-analyzed the images using total integrated Green/Red signal as a ratiometric readout as suggested. However, this analysis did not provide the same quality and consistency as our established pipeline. We complemented the imaging data with a more representative immunoblot which is now included in Figure 3 D.

- Fig. S4B –shouldn't there be any mCh⁺/GFP⁻ dots, because Baf should prevent the acidification of lysosomes? What about WB with +/- Baf in addition?

We repeated the experiments and reanalyzed the data which are now shown in Figure S4 C. The new results further validate that Bafilomycin A1 inhibits lysosomal acidification and formation of mCh⁺/GFP⁻ punctae. As mentioned above, we applied Bafilomycin A1 only for 15 h due to cell toxicity and were not able to look at autophagic degradation by immunoblot.

- Text 168-170: Where can one see the large structures? In general, more frequent indications where to find the respective data would have been helpful.

We thank reviewer #3 for raising this point. We recognize that we did not direct the reader where to find these observations the original manuscript, leading to confusion. Large mCh⁺/GFP⁺ structures are now highlighted with yellow arrows in Figure S5 A, Figure S5 C, Figure S6 A and Figure S6 E.

- Text 171-173 / Fig 3F: Δ PB1 mutant was significantly less potent in clearing (50%) and not "almost as effective". Needs to be corrected. The fact that the activity of the point mutation is higher than Δ PB1 mutant may indicate remaining oligomerization properties and that probably endogenous p62 is contributing a lot to the clearance. KD of endogenous p62 needed to be able to evaluate this, because the LIR is probably as important and it is mostly/only endogenous p62-LIR that preserves functionality of this mutation. Since the authors now do have a p62 KO cell line, this specific experiment could be reproduced under clean p62 KO conditions.

We corrected the wording in the manuscript to reflect that the Δ PB1 mutant was less potent in clearing targets as opposed to being "almost as effective." This adjustment ensures that the text accurately represents the experimental data. Our data in Figure S9 D indicate that the p62 K7A/D69A mutant retains some ability to self-oligomerize, albeit to a lesser degree than wild type p62. This is consistent with the observed differences in functionality between the Δ PB1 mutant and the point mutant.

We want to thank reviewer #3 for suggesting performing experiments to evaluate the contribution of endogenous p62. We generated HeLa mito-mCh-GFP p62 K.O. cell lines and performed additional experiments using VHH_{GFP}-p62 mutants (Figure S6). These experiments provide insights into the impact of endogenous p62 on the functionality of the mutants and are described in lines 182ff and 205ff.

- Fig. 4G: There seems to be a quite high background of binding to beads. What did the authors consider as robust enrichment and why not plotting IP WT/KO against each other? That should give a more specific picture.

Compare response to major point #5 of reviewer #2. Lysates from Huh7 and Huh7 p62 K.O. cells were processed in separate MS runs making a differential analysis between these two experiments not reliable.

- Text 250ff – is the effect seen rather sth similar to a Hook effect than preventing oligomerization? Too much dilution by forming individual dimers rather than oligomers? There is literature out there that PB1 and UBA domain are contributing to the formation of larger oligomers. Using a construct that is not limited to ZZ

domain but includes the PB1 domain in Fig. 4C could further clarify this point as well.

Compare response to major point #1 or reviewer #1. Testing recombinant PB1 via SPR is challenging due to its tendency to aggregate. Since the p62 immunogen in our Llama immunization campaign lacked the PB1 domain, it is highly unlikely that VHH_{p62} recognizes the PB1 domain via a high affinity interaction. We adjusted the text and Figure 4 A to clarify that the immunogen did not contain the PB1 domain.

Other minor points:

- Fig 3D: why does the mCh signal disappears while GFP signal partly stays? Really from the same blot or potentially two different ones with issues in sample transfer?

This difference likely originated from inconsistent sample transfer. We re-ran the samples and included a new Western blot image in Figure 3D.

- Transient and stable expression should be indicated/defined for each construct in all figure legends.
- The terms “selective” and “potent” should be used carefully according to their actual definition.

Thank you. We corrected the text accordingly.

22.9.2025

Beat Nyfeler and Zacharias Thiel

Subject : Nature Communications Manuscript NCOMMS-24-68163-T

REVIEWERS' COMMENTS

Reviewer #1 (Remarks to the Author):

The authors have carried out extensive experiments, and it is highly commendable that they demonstrated the degrader can also break down multimeric reporter proteins. Overall, the manuscript has been substantially strengthened.

We would like to thank reviewer #1.

Reviewer #2 (Remarks to the Author):

The authors have adequately addressed all of our concerns through both additional experiments and explanatory comments. They have provided new data that strengthen their conclusions as well as reasonable explanations. However, the authors only included quantification of microscopic images in Figures S2B, S2C, S4C, and S10B. They should also include representative images corresponding to these quantifications.

We would like to thank reviewer #2. We have included representative images in figures S2, S4, and S10 in the new version of the manuscript.

Reviewer #3 (Remarks to the Author):

The authors have addressed my concerns sufficiently to recommend publication in Nature Communications. I congratulate them on this nice piece of work.

We would like to thank reviewer #3.

Reviewer #4 (Remarks to the Author):

We would like to thank reviewer #4.